# The feasibility of reaching gigatonne scale CO$_2$ storage by mid-century

Yuting Zhang[1] ✉, Christopher Jackson[1] & Samuel Krevor [1]

The Sixth Assessment Report by the Intergovernmental Panel on Climate Change projects subsurface carbon storage at rates of 1 – 30 GtCO$_2$ yr$^{-1}$ by 2050. These projections, however, overlook potential geological, geographical, and techno-economic limitations to growth. We evaluate the feasibility of scaling up CO$_2$ storage using a geographically resolved growth model that considers constraints from both geology and scale-up rate. Our results suggest a maximum global storage rate of 16 GtCO$_2$ yr$^{-1}$ by 2050, but this is contingent on the United States contributing 60% of the total. These values contrast with projections in the Sixth Assessment Report that vastly overestimate the feasibility of deployment in China, Indonesia, and South Korea. A feasible benchmark for global CO$_2$ storage projections, and consistent with current government technology roadmaps, suggests a global storage rate of 5-6 GtCO$_2$ yr$^{-1}$, with the United States contributing around 1 GtCO$_2$ yr$^{-1}$.

Climate change mitigation scenarios limiting global warming to <2 °C, such as those compiled by the Intergovernmental Panel on Climate Change (IPCC), consistently anticipate widespread adoption of geological (subsurface) CO$_2$ storage globally[1-13]. By mid-century, the interquartile range of annual injection rates of CO$_2$ in the scenarios in the IPCC's Sixth Assessment Report is more than 6 GtCO$_2$ yr$^{-1}$ (the full range is 1–30 GtCO$_2$ yr$^{-1}$)[3]. These projections of CO$_2$ storage deployment arise from integrated assessment models (IAMs), which are tools to evaluate self-consistent transformation pathways of the global economy-energy-emissions system[6]. The envisaged CO$_2$ storage industry is comparable to the current scale of the hydrocarbon industry. Globally, 4 Gt of oil was produced annually between 2011–2021[14].

Projections of the rapid ramp-up of CO$_2$ storage in integrated assessment models depend on assumptions that geological CO$_2$ storage resources are sufficiently ubiquitous[15,16], and that carbon capture and storage (CCS) can be used for emissions mitigation across numerous sectors[17-20]. The perspective that regional storage use is uninhibited by geological factors is usually justified by estimates of storage resources that range between 10,000 – 30,000 Gt globally[15,21-27]. For certain industrial sectors, CCS is considered the only mitigation option available to reduce emissions[28]. As a result, integrated assessment models that have an embedded assumption of a global storage resource base >5000 Gt, are generating scenarios that rely on a maximum theoretical potential of CO$_2$ storage[29].

The deployment of CCS has fallen short of near-term projections from integrated assessment models[30-36]. Globally, ~70% of the 149 projects proposed to be operational by 2020, aiming to store 130 Mt of CO$_2$ annually, were not implemented[37]. Project cost, low technology readiness levels among the capture technology, and a lack of revenue streams, e.g., oil production, are among the main contributors to projects stopping[37,38]. Among existing actively operational CCS projects, only around 9 Mt yr$^{-1}$ of a total capture capacity of 45 Mt yr$^{-1}$ is injected for dedicated storage, with the rest used for enhanced oil recovery[39]. This discrepancy between real-world development and projected trajectories highlights limitations to CO$_2$ storage scale-up that are not captured by existing integrated assessment models.

Many potentially leading geographic, geologic, and techno-economic limitations to subsurface resource exploitation are not yet represented in integrated assessment models[16,29,40-44]. Integrated assessment models use a range of constraints on the scale-up of CO$_2$ storage (see Fig. 1. For full descriptions of the integrated assessment models included in the figure, see refs. 1,2,4,7–9,11–13). These include single values of global storage potential, supply cost curves and limits on injection rates, but some also have no subsurface-specific constraints. However, these constraints have no direct correlation to storage resource deployment in the resulting trajectories. Figure 1 shows that the use of cost supply curves, the most granular representation of subsurface storage in integrated assessment models, can

[1]Department of Earth Science and Engineering, Imperial College London, Exhibition Road, London, UK. ✉e-mail: yuting.zhang16@imperial.ac.uk

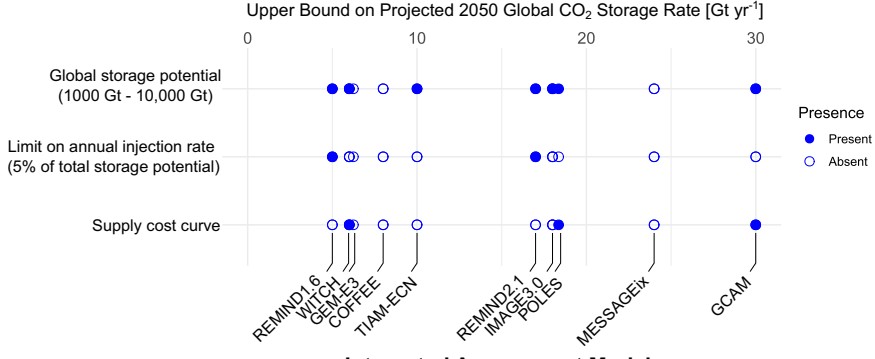

**Fig. 1 | Comparative review of constraints on CO₂ storage deployment across integrated assessment models.** Plot showing the presence and absence of constraints employed in the representation of CO₂ storage deployment across IAMs that were used in the AR6 projections. The horizontal distribution of the models indicates the upper bound on projected CO₂ storage rate. The global storage potential range between 1000 Gt (REMIND1.6 and TIAM-ECN model) to more than 10,000 Gt (WITCH model) across integrated assessment models[1,2,4,7–9,11–13]. REMIND1.6 and REMIND2.1 employed a limitation on annual injection rate of 5 Gt yr⁻¹ and 20 Gt yr⁻¹, respectively[1,9]. No constraints applied specifically to the representation of CO₂ storage have been identified for GEM-E3, COFFEE, and MESSAGEix[4,7,8]. Overall, these integrated assessment models represent 82% of total projections (689 modelled outcomes) compiled across the four climate categories we analyse within the AR6 database.

lead to both the largest and among the smallest scales of deployment in the projections.

The evaluation of CO₂ storage scale-up by using more restrictive storage capacities or by direct comparison to industrial analogues reveals significant global and regional discrepancies from the projections of conventional integrated assessment models[29,42–47]. The fundamental flow physics of CO₂ migration and trapping have been characterised in laboratory studies[48–50]. These properties have been incorporated into physics-based models to analyse limitations to large-scale CCS deployment arising from injectivity and plume migration in specific regions[42–44,51–53]. The use of historical hydrocarbon production rates, and the drilling of wells, as a proxy for CO₂ storage scale-up shows that historical rates of engineering are similar to projections in the USA, whereas there is less precedent in Asian countries, and particularly China[29,46,47]. Embedding these empirical restrictions into integrated assessment models significantly restricts the deployment of CCS[29,45]. However, this direct use of historical analogue data is difficult to generalise as a modelling approach because of challenges in translating the original datasets to units and processes associated with subsurface CO₂ storage, and the limited flexibility in the exploration of uncertainties in deployment trajectories.

Growth modelling frameworks, of which the logistic curve is the most widely used mathematical form, have been developed expressly to create future projections of natural resource consumption, based on observed patterns of growth in extractive industries[54–60]. The application of a growth model to the scale-up of CO₂ storage globally, without resolving regional variations, suggested deployment of no more than 11 Gt by 2050, but with many hundreds of Gt potentially stored by the end of the century[61]. This approach has also been used at the regional scale, identifying boundaries for plausible scale-up trajectories in the USA and Europe[62,63]. The strength of the logistic model is derived from a combination of its simplicity, its embedding of the impact of the depletable nature of CO₂ storage resources on growth, and the ability to make use of growth parameterisations based on historical analogues from extractive and other industries[64]. The important trade-offs come from its lack of granularity. The framework does not explicitly define particular incentives or inhibitions to growth arising from engineering, geology, and economics, and it is only valid for evaluating the growth of a large number of deployments over at least multiple decades[54,65–67].

Here, we generate projections of CO₂ storage deployment geospatially around the world that are constrained by current or planned deployment, rates of growth, and the assessed geological resource base. We use a growth modelling framework employing a symmetric logistic curve, building directly on the work analyses of subsurface CO₂ storage in the USA and Europe from Zhang et al.[62] and Zhang et al.[63]. We expand this to geographically resolved models of scale-up across North America, Europe, the Middle East, and Asia. We account for current CCS deployment and the geological constraints in each region by anchoring the modelled trajectories using the cumulative storage anticipated by 2030, and the assessed resource base. We formulate six scenarios under this framework, introducing variations in storage resource constraints, upper bounds on growth rates, and targeted regional limits on storage rate, to examine the uncertainty and possibilities surrounding projected CO₂ storage development (Table 1). We model a distribution of storage rates for each region by selecting trajectories randomly within a bounding parameter space of feasible annual growth rates and the assessed storage resource base. The distribution of global storage rates is then the sum of the modelled storage contributions from each region. We use these projections to identify a geography of feasible CO₂ storage scale-up, to analyse the trajectories in integrated assessment models, and identify the impacts of regional contributions in driving global storage rates.

## Results

### A map of geographically resolved CO₂ storage scale-up

We generate model projections for storage development across ten regions identified because they currently have some CO₂ storage activity (Fig. 2). These are the base results from which we derive our global scenarios. Each point on a graph in Fig. 2 shows a combination of growth rate and storage resource base that parameterises a scale-up trajectory for that region (See methods). To consider the scenarios summarised in Table 1, we select from points that fall within areas of the graphs corresponding to constraints placed on both maximum storage resource (horizontal lines) and maximum growth rate (vertical lines). These points parameterise the scale-up trajectories in our model subject to the constraints. We now analyse the implications of these regional projections for global scale-up by combining and filtering these results to explore scenarios of interest.

### The feasibility of global CO₂ storage projections in integrated assessment models

We first use our regional projections to evaluate the feasibility of global CO₂ storage deployed in the results compiled in the IPCC Sixth

**Table 1 | Summary of constraints in the modelling of trajectories of CO$_2$ storage for six proposed scenarios**

| Scenario | Description of constraints | | |
| --- | --- | --- | --- |
| | Upper bound of storage resource availability | Upper bound of growth rate | Storage rate in 2050 |
| Reference | Central estimates representing current estimates from existing national and regional storage resource assessments | 20% for all regions except 25% for China | Unconstrained |
| Minimum | Hypothetical minimum that is one order of magnitude lower than current estimates | 10% for all regions | Unconstrained |
| Maximum | Hypothetical maximum that is one order of magnitude higher than current estimates | 20% for all regions except 25% for China | Unconstrained |
| Growth10% | Central estimates, i.e., same as the Reference scenario | 10% for all regions | Unconstrained |
| US1Gt | Central estimates, i.e., same as the Reference scenario | 20% for all regions except 25% for China | Limited to 1.04 Gt yr$^{-1}$ in the USA |
| EUUSChina | Central estimates, i.e., same as the Reference scenario | 20% for all regions except 25% for China | Limited to 1.04 Gt yr$^{-1}$ in the USA, 0.33 Gt yr$^{-1}$ in the EU, 0.175 Gt yr$^{-1}$ in the UK and 0.216 Gt yr$^{-1}$ in China |

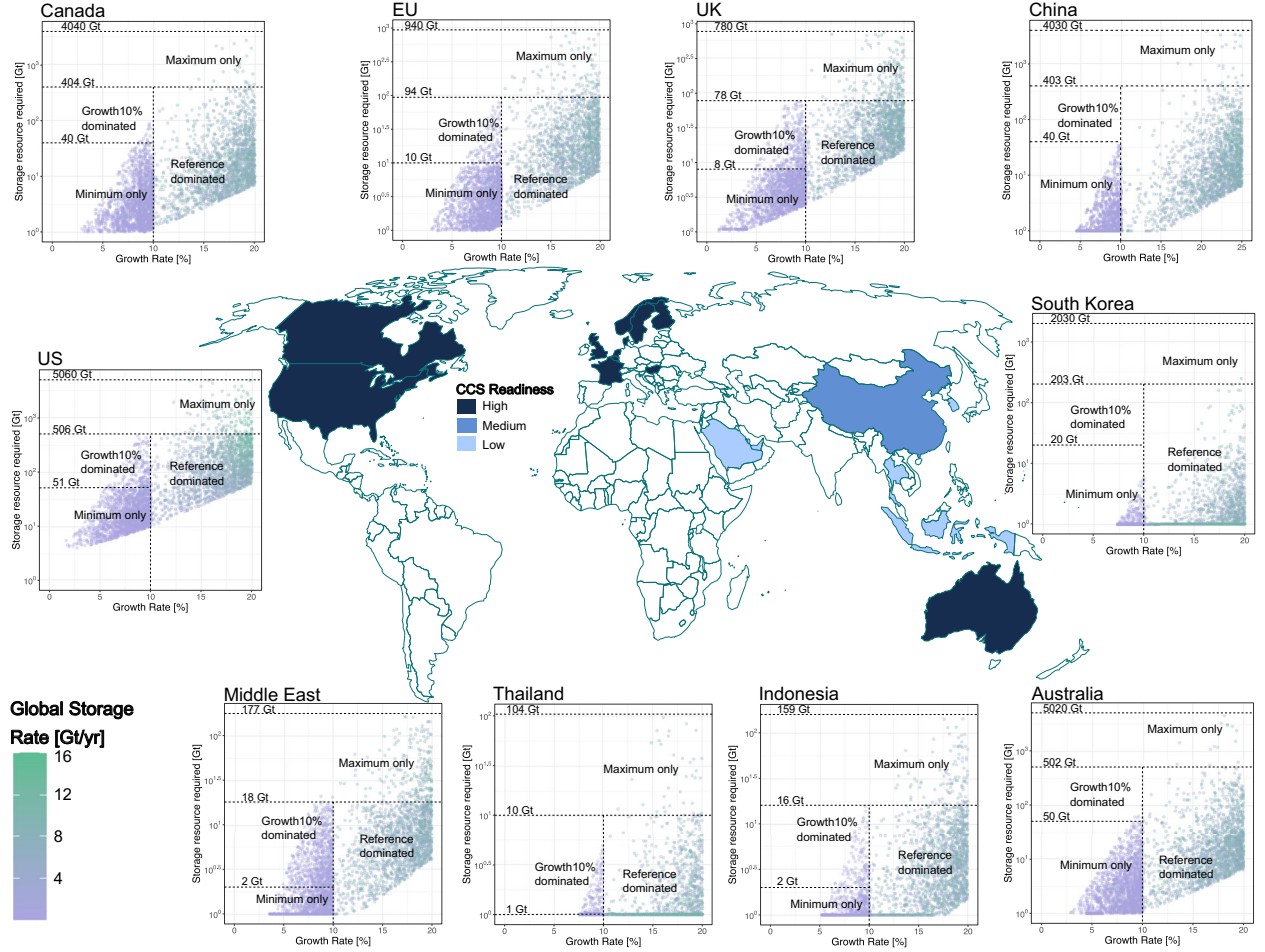

**Fig. 2 | Schematic overview of geographically resolved CO$_2$ storage scale-up projections.** Schematic map of global storage regions each associated with a degree of CCS readiness to commercialise large-scale CO$_2$ storage (See methods). The points in the graphs each represent the parameterisation of a modelled scale-up trajectory, parameterised by the growth rate and storage resource, and within regions of the graph corresponding to the Reference (constrained by central estimates of storage resource bases and up to 20% of growth rate for all regions except 25% for China), Minimum (constrained by a resource base that is 10% of central estimates and up to 10% of growth rate for all regions), Maximum (constrained by a resource base that is an order of magnitude higher than central estimates and up to 20% of growth rate for all regions except 25% for China) and Growth10% scenarios (constrained by central estimates of storage resource bases and a growth rate of 10%; see Table 1). The colour of each marker shows the combined global storage rate to which this trajectory contributes. Here the graphs are shown as a part of a schematic to illustrate the entirety of the modelling results. The full-size plot of each region is available in Supplementary Fig. 1.

Assessment Report (AR6)[3,6]. Figure 3 shows boxplots of annual projected CO$_2$ storage rate in 2050 for four climate categories of the AR6, including pathways limiting warming to 1.5 °C with limited (red) or high overshoot (green), and 2 °C with a likelihood of >50% (purple) and >67% (blue). We overlie the projected storage rates from the AR6 with six ensembles of projected CO$_2$ storage rates obtained from our growth modelling framework. Each distribution represents the results of the growth modelling subject to different

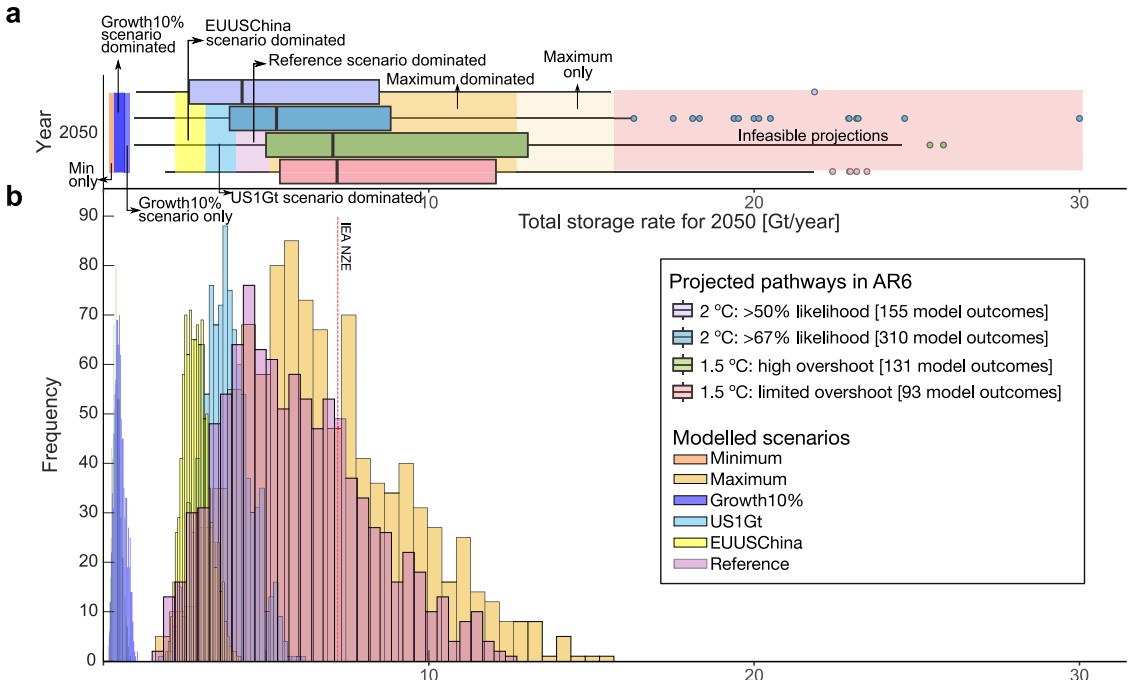

**Fig. 3 | Comparison of projections of feasible global aggregate storage rates across six scenarios with projected global CO$_2$ storage rates from integrated assessment models. a** Above the horizontal axis shows horizontal boxplots of global projected storage rate in Gt per year based on 465 model outcomes on limiting warming to 2 °C with a probability of >50% (purple) and >67% (blue) and 224 model outcomes on limiting warming within 1.5 °C with high overshoot (green) or limited (red). **b** Below the horizontal axis shows the histograms of modelled distribution of globe aggregate storage rate across ten regions under six different scenarios using the logistic modelling framework. The description of each scenario is provided in Table 1.

groups of constraints on storage resource availability, maximum annual growth rate and limits on deployment in specific countries (Table 1).

The comparison reveals several limitations in the projections from integrated assessment models. Around 8% of the trajectories from the AR6 (56 out of a total of 689 model outcomes) project rates of storage in 2050 that are greater than 16 Gt yr$^{-1}$. Our analysis identifies these pathways as infeasible, requiring sustained annual growth in excess of 20% and storage resources in excess of the theoretical maximum that has been evaluated for individual countries (see Maximum scenario, Table 1). The interquartile range of projected pathways for both 1.5 and 2 °C (>67% likelihood), and the IEA Net Zero Emission target[5] – widely considered as a benchmark to decarbonise the global energy sector – are achievable by storage scale-up modelled largely in the Reference and the Maximum scenarios. In contrast, limiting sustained annual growth to <10%, a rate still greater than what has been achieved in the past 20 years in the CCS industry[35], inhibits the attainable aggregate global storage rate to a maximum of 1 Gt yr$^{-1}$, below any projections of storage deployment in the 1.5 and 2 °C pathways of the AR6.

Limiting regional deployment based on current government roadmaps has major implications for the global total. The USA has identified rates of 1 Gt yr$^{-1}$ of geological storage by 2050 in its Long-Term Strategy[68]. If this is considered a maximum for the USA in 2050 (US1Gt, Table 1), then the global aggregate storage rate cannot exceed 6 Gt yr$^{-1}$ in 2050. Note that the maximal case in the maximum scenario, with a feasible global storage rate of 16 Gt yr$^{-1}$ is also contingent on the USA contributing at least 10 Gt yr$^{-1}$ or 60% of the total. In another scenario (EUUSChina, Table 1), we have limited deployment in the UK and the EU to their stated storage targets[10,69], and deployment in China 0.216 Gt CO$_2$ yr$^{-1}$, the mass rate of the volumetric equivalent of the maximum historical production rate of oil[14]. Under these constraints, the global aggregate storage rate is limited to a maximum of 5 Gt yr$^{-1}$.

## Global storage rates are driven by storage rates achieved in six regions

Figure 4 displays the modelled storage rates across the ten storage regions for the Reference scenario. We compare these modelled storage rates with the range of projected CO$_2$ storage rates in Australia, Canada, USA, China, Indonesia, South Korea as presented in the AR6 (indicated by a red square bar in Fig. 4). Additionally, we compare the feasible distribution of modelled storage rate targets with storage targets published in government decarbonisation roadmaps for the UK, EU, and the USA (indicated by a blue square in Fig. 4). No projected storage rates, or national targets have been identified for Thailand in the IPCC database or national reports. Projected storage rates within integrated assessment models for the EU and Middle East are reported in aggregate and cover different geographical regions and are not directly comparable with the storage rates derived from our models.

Across the ten regions, IPCC-compiled projections of CO$_2$ storage exceed our upper bounds of feasibility (Reference scenario) for Asian countries, i.e., China, Indonesia and South Korea. For instance, integrated assessment model projections for China reach up to 6.7 GtCO$_2$ yr$^{-1}$; we identify this as infeasible with our modelling framework because it requires annual growth in storage rates of >30% sustained for at least 20 years. In contrast, projected storage rates in integrated assessment models outlined for Western developed nations, i.e., Australia, Canada, and the USA are evidently more conservative and within the feasibility range of our modelled storage rates. Moreover, published CO$_2$ storage targets from announced decarbonisation roadmaps for the EU[10], UK[69] and the USA[68] also align well with the range of feasible storage rates modelled in the Reference scenario.

The relative importance of a single country or region to achieving a global storage rate can be inferred by evaluating exemplary cases. From the distribution of feasible storage rates in the Reference scenario, we show two exemplary combinations of CO$_2$ storage scale-up

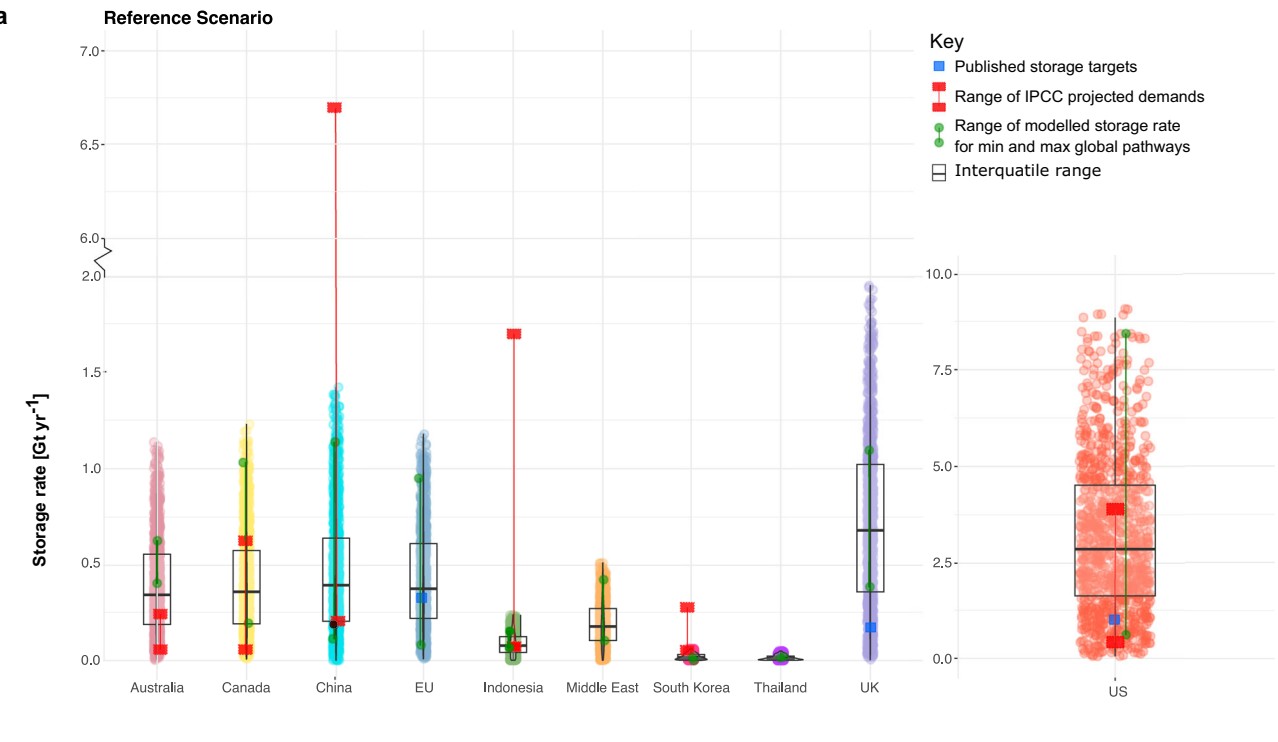

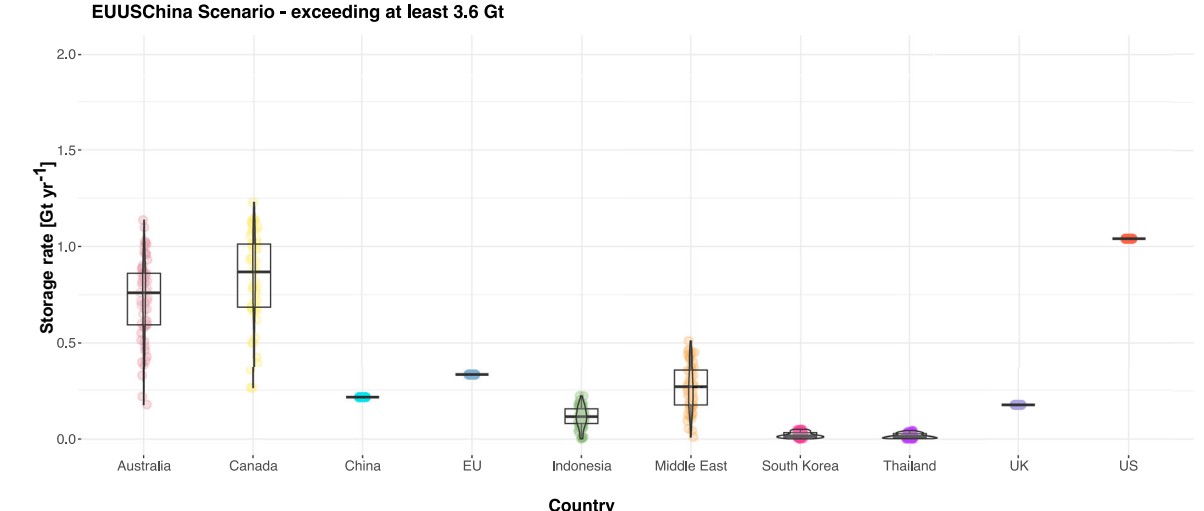

**Fig. 4 | Modelled storage rates under Reference and EUUSChina Scenarios.** Boxplots of modelled storage rate under the **a** Reference and **b** EUUSChina scenario across ten selected regions. The box represents the interquartile range, with the median indicated by the bold line. Note that plots in **a** both depict modelled storage rate for the reference scenario albeit with different scales on the y-axis between the USA and other regions. The whiskers extending from the box towards the bottom and the top represents the minimum and maximum values of modelled storage rate, respectively. In the Reference scenario, the red squares joined by a red line indicate the upper and lower bound of projected national storage rates across the regions that result in the minimum (2 Gt yr⁻¹) and maximum

compiled in the AR6[3]. The blue square for the EU, UK and the USA in the Reference scenario represents published storage targets from existing decarbonisation strategies[10,68,69]. No projected storage demands have been identified for Thailand. The upper bound of green dots across regions indicate a combination of national/regional storage rate that contribute to achieving the maximum global storage rate of 13 Gt yr⁻¹ in the Reference scenario. The lower bound of green dots illustrate a selection of modelled storage rate across regions that collectively attain the minimum global storage rate of 2 Gt yr⁻¹. Raw data is provided in the Source Data file.

across the regions that result in the minimum (2 Gt yr⁻¹) and maximum (13 Gt yr⁻¹) aggregate global storage rate, illustrated by the green dot bar in Fig. 4. The Minimum and Maximum scenarios for Thailand, South Korea and Indonesia have similar modelled storage rates (overlapping green dots). This indicates that the aggregate global storage rate is insensitive to their contribution. In contrast, the USA, China, UK, EU, Canada and the Middle East all have a large range between the exemplary minimum (2 Gt) and maximum rates (13 Gt; green dot bar in Fig. 4). The total global rate is sensitive to the contribution from these six regions.

Similarly, the impact of individual country constraints driven by policy or historical analogue can be observed. Under the EUUSChina scenario, we show the combined impact of limiting storage potential to announced strategies for the EU (0.33 Gt yr⁻¹), UK (0.175 Gt yr⁻¹), USA (1.04 Gt yr⁻¹) and historical oil production in China (0.216 Gt yr⁻¹) where there is not a published target[10,14,68,69]. We have already discussed how this scenario limits global mid-century storage rates to less than 5 Gt yr⁻¹. We now look at the implications in achieving a minimum global storage rate under this scenario of required contributions from individual regions.

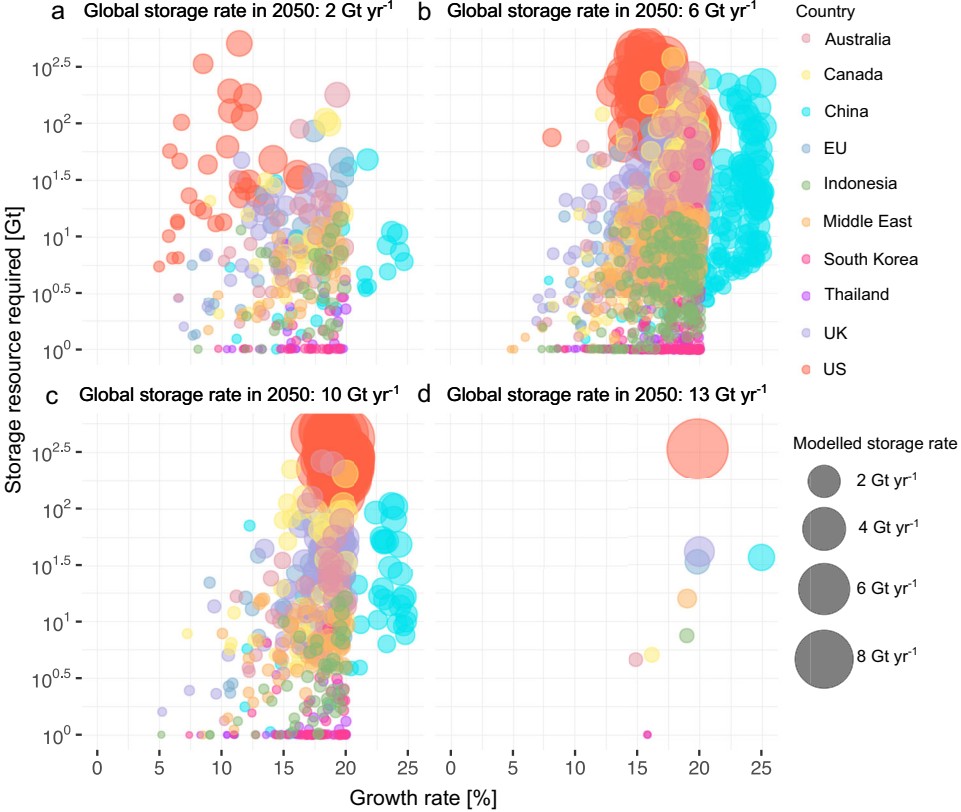

**Fig. 5 | Distribution shifts in modelled scale-up trajectories as a function of global storage rates in the Reference scenario.** Changes in the distribution of modelled scale-up trajectories with incremental increases in global storage rate from **a** 2 Gt yr$^{-1}$ to **b** 6 Gt yr$^{-1}$, **c** 10 Gt yr$^{-1}$, and **d** 13 Gt yr$^{-1}$. The colour differentiates each storage region while the size of the point represents modelled storage rate.

The distribution of storage rates by region, where the global total exceeds 3.6 Gt yr$^{-1}$ in (the lower quartile of the 2 °C pathways in the AR6) is shown in Fig. 4. In this case, the feasibility of the global aggregate storage rate depends on Australia and Canada each providing at least 0.2 Gt yr$^{-1}$ by 2050, a factor of 10 scale-up from the planned rates for 2030 in both countries. Meanwhile, for Indonesia, South Korea, Thailand and the Middle East, there is no minimum contribution required to achieve a global rate of 3.6 Gt yr$^{-1}$, but the lower the storage rate in those regions, the greater the required contribution from Australia and Canada to achieve the global total.

**The geography of possible CO$_2$ storage deployment changes with global rate targets**

Here, we show how the distribution of storage rates for each region evolves as a minimum global storage rate is varied for the Reference scenario (Fig. 5). At a low global storage rate of 2 Gt yr$^{-1}$, there are a range of feasible scale-up trajectories where the Western developed nations (Australia, Canada, the UK and EU) make similar contributions towards the total. However, as the global storage rate is increased, the USA must increasingly take on a large fraction of the total. The USA must store nearly 70% (8.4 Gt yr$^{-1}$) of the total when the highest global rate of the Reference scenario, 13 Gt yr$^{-1}$, is achieved. The density of modelled storage trajectories for the USA moves progressively towards the upper right corner of the parameter space, characterised by sustained high growth rates of 18–20% and a dependency on a storage resource base that is as large as the central estimates of 506 Gt. In China, even a relatively small contribution to the global total requires extraordinary growth, above 20% sustained to 2050. The decreasing number of points with increasing global storage target shows the narrowing of possible individual region trajectories as the global total increases.

## Discussion

We evaluate feasible global CO$_2$ storage rates for 2050, combining contributions across 10 geographic regions, which have currently active or planned CCS deployment. There is a maximum feasible combined CO$_2$ storage rate of 16 Gt yr$^{-1}$ by 2050, encompassing 92% of the 689 projections of scale-up in the 1.5 and 2 °C climate categories of the Sixth Assessment Report of the IPCC. The feasible limit reduces to 13 Gt yr$^{-1}$ for our reference scenario where the central estimates of storage resources are used. However, if deployment in the USA is limited to projections in government roadmaps, or indeed central projections from pathways in the AR6, the global deployment of CCS is further limited to no more than 6 Gt yr$^{-1}$. In the most conservative scenarios we explored, limiting the annual growth rate to less than 10% and the storage resource base to 10% of the current estimates reduces the attainable global storage rate to less than 1 Gty$^{-1}$.

The feasible regional distribution of CO$_2$ storage deployment varies considerably from the projections emergent from integrated assessment models. The maximum rate of storage achieved globally is largely controlled by deployment in six regions, the USA, China, the UK, EU, Canada and the Middle East, in order of decreasing impact. Projections from integrated assessment models envision global mitigation from CCS by mid-century in excess of 10 Gt yr$^{-1}$ with relatively modest contributions from the USA. However, these projections vastly overestimate feasible deployment in China, Indonesia and South Korea. As described above, when limiting 2050 deployment in the USA to 1 Gt yr$^{-1}$, as per projections in both government roadmaps and from integrated assessment models, the global rate achieved in 2050 cannot exceed 6 Gt yr$^{-1}$. This further reduces to 5 Gt yr$^{-1}$ when government projections are also used for the EU (0.33 Gt yr$^{-1}$), UK (0.175 Gt yr$^{-1}$) and deployment in China is limited to, 0.216 Gt yr$^{-1}$, the volume equivalent of historical oil production[14].

These findings are remarkably consistent with the analyses that impose empirical constraints directly from historical oil and gas production, and also find a global annual injection rate of $CO_2$ limited to $5-6$ Gt yr$^{-1}$, with the USA as the largest contributor to that total[29]. The convergence of these estimates from distinct modelling approaches suggests that a benchmark for the upper bounds of the feasible global deployment of CCS by 2050 could be set at $5-6$ GtCO$_2$ yr$^{-1}$.

While technically feasible, the range of projections between $6-16$ GtCO$_2$ yr$^{-1}$, representing a large proportion of the AR6 pathways, are fundamentally more uncertain than lower projections because of the absence of any current business or political framework under which this scale of industry would operate. In these scenarios the USA would have to store a majority percentage of its current $CO_2$ emissions, likely requiring major cross-border transport of $CO_2$. International agreements for the disposal of $CO_2$ are emerging in Europe and Asia[70]. However, plans that would incentivise such a large-scale deployment in North America are not currently in development.

Projections from integrated assessment models should revisit and add to the constraints that reflect limitations arising from subsurface storage development. Our results suggest that gigatonne scale mitigation from $CO_2$ storage is feasible, but with deployment trajectories and geographies differing from current projections emergent from integrated assessment models. The Sixth Assessment Report includes unrealistic projections of CCS deployment, particularly in Asian countries, and these are widely used as benchmarks for progress measurement.

This work demonstrates the practicality of using available data on injection rates, subsurface storage resources, and a logistic growth modelling framework to provide multi-decadal constraints on the pace of future growth in $CO_2$ storage. Their simple parameterisation permits the rapid assessment of the impacts of uncertainties in the capacity to realise planned developments, to incentivise growth, and in the understanding of the underlying storage resource base. Growth models do not have high temporal fidelity, they represent a statistical averaging over yearly fluctuations in development. An obvious next stage in development would be to combine the multi-decadal constraints from growth models with the more granular cost supply curves and physics-based models of injectivity that can provide deterministic estimates of yearly fluctuations in development.

## Methods

### Selection of storage regions

For our study, we only consider those regions that have existing or planned CCS projects with announced capture capacities published in the 2022 Global Status report by GCCSI that will reach an operational date prior to or during 2030[71]. This narrows down our consideration to ten regions, encompassing countries including Australia, Canada, China, Indonesia, South Korea, Thailand, the UK and the USA, along with the EU and the Middle East. For the consideration of storage resource availability in the EU, we constrained the resource base to the estimated offshore asset in the Norwegian North Sea. This is a simplified consideration given that the potential to develop onshore storage resources across continental Europe faces opposition and uncertainty[72-76]. For our consideration of countries that could contribute towards CCS within the EU region by 2030, we consolidated capacity from capture plants where announced, i.e., France, Belgium, Norway, Finland, Sweden, Hungry, the Netherlands and Denmark (8 out of the 27 EU countries). While there are more EU countries that have announced projects, i.e., Italy and Ireland[71], their annual capture capacity is yet to be determined and they are thus excluded. For the Middle East, we aggregated Saudi Arabia, Qatar and United Arab Emirates, which have operational commercial-scale CCS plants and published plans for future developments.

Finally, we differentiated our ten selected regions under three levels of readiness: high, medium and low. For this, we make use of the 2018 Global CCS Readiness Index Report (GCCSI-RI)[77] which scores each nation using a series of criteria categorised under four indicators: inherent interest of CCS which represents a nation's dependence on hydrocarbon products, legal regime, policy measures that support CCS, and the level of maturity of storage resource assessment. A high level of CCS readiness in our study corresponds to regions scoring between $60-80$ in the GCCSI-RI. A medium level indicates those regions where there is generally limited policy and regulatory support, but ongoing projects exists (scores $40-59$ in the GCCSI-RI). Lastly, a low level of readiness is associated with regions that have limited demonstration of $CO_2$ storage, a lack of adequate policy framework in support, and the volumetric estimate of storage resource base across oil and gas, and saline aquifers largely remain unknown (scores $20-39$ in the GCCSI-RI).

### Carbon dioxide storage projections in integrated assessment models

Integrated assessment models are analytical frameworks capturing key interactions of the human-earth system to understand implications across disciplines of energy, economy and the environment simultaneously[78]. For climate change mitigation, integrated assessment models are employed to project greenhouse gas emission reductions that achieve long-term climate objectives (for 2100) or policy goals (generally for 2050) whilst exploring different cost-effective strategies to decarbonise the energy supply[79].

The AR6 Scenario Explorer is an online platform hosted by the International Institute for Applied Systems Analysis (IIASA) that provides a comprehensive compilation of decarbonisation pathways generated by various integrated assessment models underpinning the Sixth Assessment Report (AR6) of Working Group III by the IPCC[3,6]. We make use of this database to select scenarios of $CO_2$ storage deployment at the global level that are compatible with two global warming levels, 1) limiting warming to 1.5 °C in 2100 (50% likelihood) with no or limited overshoot, and 2) limiting warming to 2 °C in 2100 (67% and 50% likelihood). We examine the key variable of total $CO_2$ sequestration through CCS and removals in Mt $CO_2$ yr$^{-1}$ for 2050, including $CO_2$ emissions captured from bioenergy use, fossil fuel use, and industrial processes. We refer to storage deployment modelled by integrated assessment models as projected scenarios.

For our analysis, we assume all $CO_2$ storage is within geological reservoirs including depleted hydrocarbon fields and saline aquifers. We do not differentiate modelled storage deployment by $CO_2$ capture source.

### Logistic growth modelling framework

We make use of a logistic growth model which is an empirical mathematical framework that has been applied to describe growth in the extractive industries, including oil and gas, coal and uranium production[54-59,80]. Over time, the model's application has been expanding beyond fossil fuel resources with several studies demonstrating the suitability of logistic curves to model energy production from renewable sources, nuclear and the rates of technological substitution[81-86].

Logistic growth models for $CO_2$ storage have been applied by the authors of this work at both regional and global scales[61-63]. In this context, the strength of the logistic growth model is to embed the impact of the depletable nature of $CO_2$ storage resources on growth, capturing the relationship between market dynamics and the physical use of the resource base. The framework does not explicitly define particular incentives or inhibitions to growth arising from policy, market demand, regulatory requirements and public acceptability[65-67]. An important trade-off relative to the more granular representations of CCS used in integrated assessment models, like cost-supply curves, is that the logistic modelling framework is only valid for evaluating growth trajectories over multiple decades or longer[54,65-67]. However,

**Table 2 | Summary of constraints employed in the logistic model**

| Regions | GCCSI CCS readiness (scored out of 100) | | Constraint on logistic model: cumulative storage in 2030 [Mt] | Constraint on logistic model: available storage resource base [Gt] | | | Constraint: growth rate parameter |
|---|---|---|---|---|---|---|---|
| | | | | Hypothetical minimum | Central | Hypothetical maximum | |
| Canada | 71 | High | 121 | 40 | 404 | 4040 | Up to 20% |
| USA | 70 | – | 1084 | 51 | 506 | 5060 | Up to 20% |
| EU + Norway | 67 | – | 129 | 10 | 94 | 940 | Up to 20% |
| UK | 65 | – | 258 | 8 | 78 | 780 | Up to 20% |
| Australia | 62 | – | 110 | 50 | 502 | 5020 | Up to 20% |
| China | 53 | Medium | 41 | 40 | 403 | 4030 | Up to 25% |
| South Korea | 37 | Low | 6 | 20 | 203 | 2034 | Up to 20% |
| Middle East | 36 | – | 71 | 2 | 18 | 177 | Up to 20% |
| Indonesia | 31 | – | 26 | 2 | 16 | 159 | Up to 20% |
| Thailand | 21 | – | 5 | 1 | 10 | 104 | Up to 20% |

Constraints include the three bounds of available storage resource base, and the region-specific cumulative storage in 2030 based on existing and planned capture capacity. Additionally, the level of CCS readiness for ten selected regions is also shown.

benchmarks from historical precedents in analogous industries allow the modelling framework to be used as a simple but robust check for the feasibility of projected storage demand modelled using more granular approaches[64].

A description of the cumulative storage, $P(t)$ [GtCO$_2$], and storage rate, $Q(t)$ [GtCO$_2$ yr$^{-1}$], of CO$_2$ at time, $t$ [yr], is outlined as Eq. (1) and Eq. (2)

$$P(t) = \frac{C}{1 + \exp\left(r\left(t_p - t\right)\right)} \tag{1}$$

$$Q(t) = \frac{C \cdot r \cdot \exp\left(r\left(t_p - t\right)\right)}{\left(1 + \exp\left(r\left(t_p - t\right)\right)\right)^2} \tag{2}$$

The logistic growth rate, $r$ [yr$^{-1}$], which hereinafter referred to as the growth rate, characterises the early part of the trajectory. This phase signifies near-exponential growth driven by expansive adoption of commercial practices, with minimal impingement from geological constraints. Subsequent to peak year, $t_p$ [yr], growth declines until the exhaustion of the storage resource base, $C$ [Gt], influenced by both geological constraints and engineering capabilities.

**Constraints on the logistic growth model**
The "Global Status of CCS 2022" report by the GCCSI provides the latest update on CCS project pipeline across the world[71]. We use the capture capacity reported for both active and planned CO$_2$ storage projects to estimate the cumulative storage of CO$_2$ in 2030 for our ten selected regions (Fig. 6 and Table 2). This serves as an external input, ensuring the consistency of modelled storage rate for 2050 with real-world CCS development up to 2030. The cumulative storage for 2030, a key constraint on the logistic model, marks a take-off point at which we begin modelling the near-exponential part of the trajectory until the inflection point on the rate curve (Fig. 6).

Assessments have been undertaken across various regions of the world to estimate the available storage resource base for CO$_2$ within geological reservoirs, including depleted hydrocarbon fields and saline aquifers[25–27,87–89]. Different concepts have been developed for storage resource assessment depending on the type of storage

medium, trapping mechanism and the geomechanical structure of the formation-seal system in saline aquifers, i.e., open, semi-open, or closed[90–92]. To define the parameter space for $C$ [Gt] in Eqs. (1) and (2) which characterises the available storage resource base, we predominantly make use of the "Storage Resource Catalogue" from the Oil and Gas Climate Initiative, an energy company-led organisation that focuses on climate change mitigation[27]. This multi-year report compiles and evaluates national and regional geologic CO$_2$ storage estimates made by geological surveys, national laboratories and other organisations, focusing on saline aquifers and depleted hydrocarbon fields, consistent with the definitions of the Storage Resource Management System[27,93]. Their report describes that the prevailing approach used to make estimates within saline aquifers across our ten selected regions is derived from the volumetric model to estimate the proportion of total pore volume that can retain CO$_2$. However, for the USA and China, the representation of storage resources we adopt is more conservative. These estimates are derived from more detailed assessments that build on the volumetric estimates and considers technical restrictions that may impede access to storage resources[26,89]. The approaches for storage resource estimated within hydrocarbon fields are more uniform across different assessments. Typically, the principle of voidage replacement is applied where the volume of CO$_2$ stored is assumed to be the equivalent volume of oil or gas produced at reservoir conditions[27]. Saline aquifers have the largest storage potential and dominate the resource inventory across our ten selected regions (85% -100% of central estimates)[23]. We refer to these assessed storage resources as our central estimates (Table 2).

There is uncertainty ranging between 1–2 orders of magnitude in storage resource estimates[26,94–96]. This geological uncertainty can only be significantly reduced with further project development, which provides more detailed reservoir characterisation, and to reduce uncertainties arising from engineering aspects of the project design. Reservoir properties characterised in the laboratory and using geophysical techniques, combined with well hydraulic tests and observations of the movement of injected CO$_2$, form the empirical basis for simulations used to forecast and history match field-scale projects. These studies can reduce the geological and engineering uncertainty pertaining to the practical use of the CO$_2$ storage resource[97]. Geological and engineering characteristics combine to govern the efficiency of the total pore volume use to store CO$_2$[90,91,95]. As a result, estimates of storage efficiency range from 0.5% - 4%[96]. Many resource base

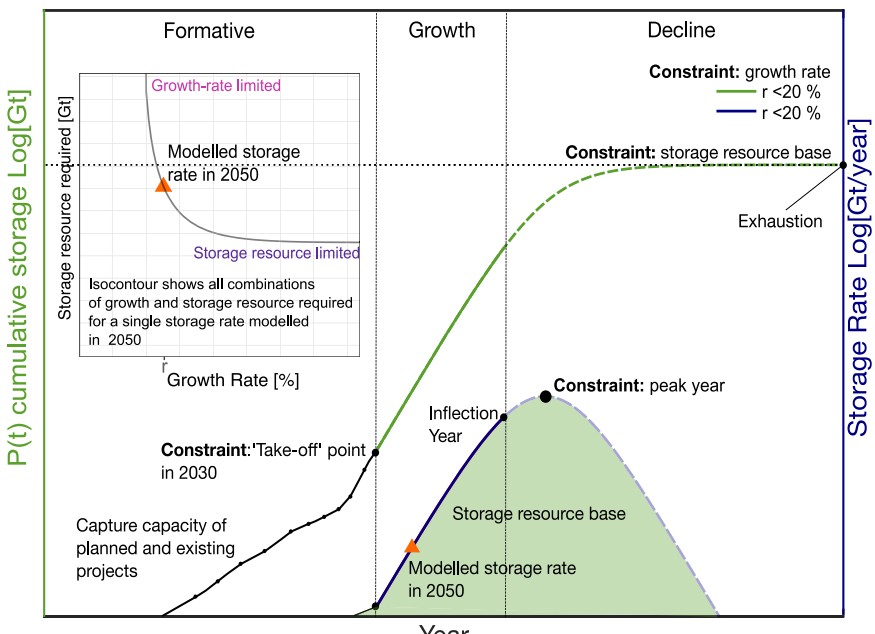

**Fig. 6 | Schematic plot of the logistic modelling framework.** Schematic plot of modelled cumulative (green curve) and storage rate (blue curve) of $CO_2$ as a function of time corresponding to Eqs. (1) and (2). Black dots connected by the smooth line are empirical data of capture capacity from existing and planned CCS projects within a given region. The 'take-off' point marks the start of an exemplary modelled growth trajectory of r%. The inset plot shows a tradeoff between storage resource and growth rate for a fixed storage rate in 2050 requiring a growth rate of r% and a certain size of a resource base. The modelled storage rate is illustrated on the rate series marked by an orange triangle. Note that these plots are for illustrative purposes; numbers are not included but the vertical axes are logarithmic, and the horizontal axis is linear.

assessments use estimates of storage efficiency obtained using reservoir simulation[87,88,98], and there is not a simple systematic relationship between dynamic and volumetric estimates of a storage resource. In light of this, we follow the approach established by Zhang et al.[62], and analyse scale-up trajectories with a storage resource base that ranges from a hypothetical minimum, 10 times lower than the central estimates of the resource base, and a hypothetical maximum, which is 10 times greater than the central estimate. In this way, our analysis implicitly incorporates uncertainties associated with the geological system, like the emergence of a limitations arising from injectivity, by the consideration of a far more restricted resource base than is currently assessed. We summarise the storage resource constraints in Table 2.

For growth rate considerations, we propose a parameter space for sustained annual growth of up to 20% for all regions except China. We consider this as technically feasible at the national or regional level, consistent with growth trajectories that are required to meet proposed $CO_2$ storage targets outlined for Europe and the USA, as well as historical precedents in growth achieved from the hydrocarbon industry[62,63]. Given the rate of acceleration observed in large-scale energy infrastructure development in China[99], we extend the maximum feasible growth rate to 25%.

Finally, a key constraint is imposed on the peak year ($t_p$) which is set to occur later than 2050 following the approach of Zhang et al.[62,63]. This reflects our main consideration for modelling scale-up trajectories of $CO_2$ storage of using the early part of the trajectory in the logistic model. Through this we are considering scenarios where the storage resource is sufficiently large that it can sustain development for timescales long enough to justify large scale investment from a business perspective. As previously mentioned, the logistic model is best considered a multi-decadal smoothed average over the typical and sometimes very large fluctuations in growth that take place in emerging industries over shorter timescales, e.g., 1–5 years.

## Modelling a storage trajectory database

To create geographically resolved $CO_2$ storage scale-up models, we compute 1000 random iterations of prospective storage rates for each region. Solutions to Eqs. (1) and (2) are numerically calculated for each given storage rate by systematically exploring all combinations within our predefined parameter space for peak year, storage resource and growth rate. We implement the solutions for the three parameters that exhibit the closest fit to our fixed input of established cumulative storage in 2030 by calculating a minimum squared difference. To generate a distribution of total aggregate storage rates, we take the sum of modelled storage rate across the ten selected regions in random combinations for 1000 iterations.

## Scenario setting

We create a number of scenarios defined by varying growth rate, storage resource and targeted limits on the regional storage rate (Fig. 7). The Reference scenario sets out a range of storage rates subject to the maximum bound we use in feasible growth rate (i.e., 20% or 25%) and central estimates of the available storage resource base for each region. The Minimum scenario reflects a conservative case, limiting the growth rate to a maximum of 10% and a storage resource base of only 10% of central estimates (hypothetical minimum). The Growth10% scenario limits growth to 10% but still uses central estimates of the storage resource base, allowing an identification of the extent to which the modelled storage rate is limited by growth versus the storage resource base. The Maximum scenario illustrates the impact of increasing the storage resource base by a factor of 10 over the central estimates. Finally, the US1Gt and EUUSChina scenarios illustrate the impact on modelled storage rate for other regions, and the global aggregate storage rate, when storage rate in the USA, the UK and the EU, are limited to announced government strategies of 1 Gt yr⁻¹ (ref. 68), 0.175 Gt yr⁻¹ (ref. 69), 0.33 Gt yr⁻¹ (ref. 10), respectively, and rates in China are 0.2 Gt yr⁻¹ (ref. 14), an estimated equivalent to historical oil production.

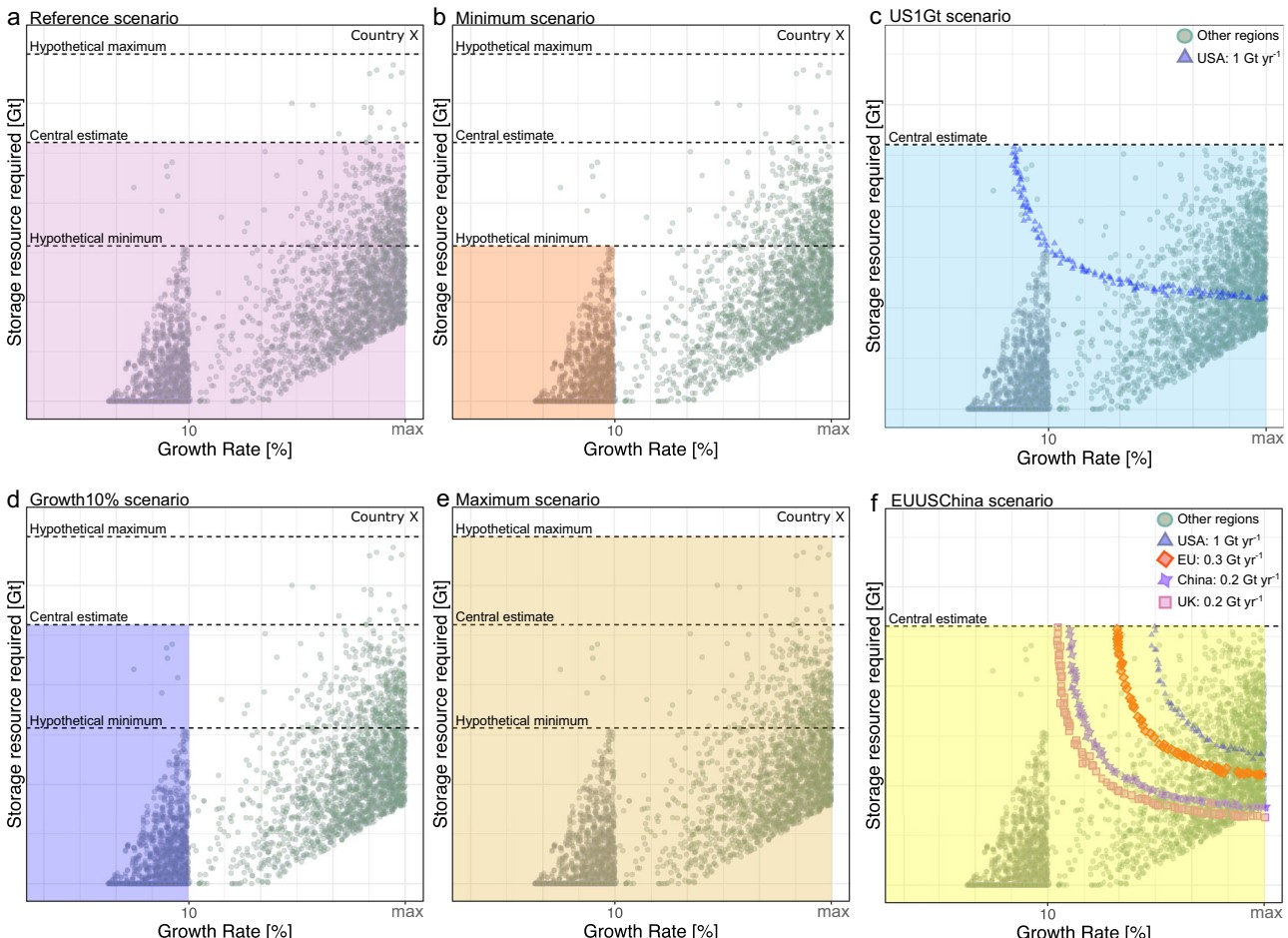

**Fig. 7 | Illustrative plots of six modelled scenarios.** The modelled scenarios are bounded by different conditions of growth rate and storage resources (coloured areas). We only show on graph for each scenario, but there will be an equivalent graph for each of the 10 countries and regions considered. **a** Reference, **b** Minimum, **d** Growth10%, and **e** Maximum scenarios explore sensitivity of modelled storage rate to changes in bounds of growth rate and storage resource estimates alone. **c** US1Gt and **f** EUUSChina scenarios explore the implication of fixed conditions on national or regional storage rate (coloured shapes) to the overall global storage prospect. Specifically, the convex loci of points in US1Gt and EUUSChina scenarios show combinations of growth and resource requirement that produces the required set target. The randomly generated storage rates for other regions are not fixed (green points).

## Data availability

The projected $CO_2$ storage rate data from various integrated assessment models for 2050 are obtained from the AR6 Scenario Explorer, which is an online platform hosted by the International Institute for Applied Systems Analysis (IIASA) at https://data.ene.iiasa.ac.at/ar6/#/login?redirect=%2Fworkspaces. The data generated in this study are provided in the Source Data file. Source data are provided in this paper.

## Code availability

The code used in this study is available in the Zenodo repository: https://doi.org/10.5281/zenodo.11446272.

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

## Acknowledgements
Funding for this work was provided by the Engineering and Physical Sciences Research Council (Grant EP/T51780X/1 to Y.Z.) and the Royal Academy of Engineering (RCSRF2223–1677 to S.K.).

## Author contributions
Y.Z. and S.K. conceived the study. Y.Z. performed the research and led the writing of the manuscript. Y.Z. developed the computer code. Y.Z., S.K. and C.J. contributed to the writing of the manuscript.

## Competing interests
The authors declare no competing interests.
