## [Peer Review File · Nature Communications]

REVIEWER COMMENTS

Reviewer #1 (Remarks to the Author):

Article presents interesting results in making projections of CO₂ storage by mid-century. It discusses the feasibility of scaling up carbon storage in subsurface reservoirs to mitigate climate change, as projected by the AR6. While the AR6 estimates a wide range of potential storage rates by 2050, this study highlights the importance of considering geological, geographical, and techno-economic limitations in these projections. Here are some comments that might improve the article.

Comments:

- 1) Line 81-83 - What does this value mean (production of CO₂ during oil and gas production?)
- 2) Line 94: some visual representation of the deviation of practical results with projections could be useful.
- 3) Line 98 - Could you please provide the reasons, why these projects were not implemented? How these factors can be taken into account while building projection models? Which factors have the highest impact?
- 4) Line 123-124 - can your projections include some practical, theoretical base (some coefficients that accounts trapping mechanisms, experimental studies - focusing on these mechanisms, hydrodynamic studies - to validate your projections? Probably, you can include some literature review of some alternatives for your approach. Is there some other research groups that are performing similar studies? what are the strong sides of your approach, and drawbacks? Adding a highlight of the novelty and global contribution in the Main part would be beneficial.
- 5) Figure 3 must be fixed - legends should be crossed by the zoomed part of the Figure (exceeding 3.6 Gt). Probably, it should be divided into a)b)c)
- 6) As authors mentioned, the capacity of potential CO₂ storage is indefinitely large value, however, not all the reservoirs/aquifers can be used for potential CO₂ storage (Line 392). Authors used reports with CO₂ storage estimates, however, it is not clear, how the selection criteria (geological and engineering constraints) of existing resources was applied in the projections. In line 407 Authors mention, that reservoir characteristics were used to test sensitivity analysis, how these results were used in the projections? Probably the description of the methodology could be extended. Also, all the constraints can be given in the Table for better visual understanding.
- 7) Conclusion might be enhanced by highlighting the contribution of this results for the industry, how it can affect the future of the industry, climate and human existence in general.

Reviewer #2 (Remarks to the Author):

Dear Editor,

This article investigates the feasibility of scaling up CO₂ storage considering geology and growth rates. Results suggest a maximum global storage rate of 16 GtCO₂ yr⁻¹ by 2050, with the USA contributing 60%. However, if constrained by government roadmaps, this drops to 5 GtCO₂ yr⁻¹, mainly due to limited deployment in the USA. Previous estimates overestimated storage in China, Indonesia, and South Korea. Achieving gigatonne scale mitigation requires updating projections to include geological and deployment rate limits.

The work is can make valuable contributions to the fields. However, I have several comments and concerns outlined below that can further improve the quality of the paper prior to acceptance.

Major Comments

1- Technical uncertainty and risk mitigation play a significant role in the estimated value. Authors are encouraged to consider those factors in their analysis. For a comprehensive list of uncertainties and risks, see the review paper recently published (<https://doi.org/10.1016/j.jgsce.2023.205117>). Without this being incorporated into the analysis, the numbers seem again overestimated (or underestimated), which weakens the contribution of this work.

2- Methods and assumptions are provided after the results and conclusions. I understand why that can be the case, but a summary of those assumptions and modeling approaches (a few lines) with proper citation and reference to the appendix must be given before discussing the results, as results make no sense to readers until they know all the assumptions made in the analysis. A slight arrangement or proper definition of models and assumptions at the end of the introduction is required.

3- With all these heterogeneous assumptions throughout the model and scenario development, how is the output of this paper any more statistically significant than simple range estimation by AR6? To be honest, I read this paper twice and was not convinced if the new modeling approach is any better or if it provided any new insights that are not already known in the field. For example, the claim “if constrained by government policies, the storage rate drops”, this is common sense! I encourage authors to morph the analysis toward revealing some aspects, like implications, that could enhance the field and better inform policymakers. Based on the abstract, the authors simply concluded for the readers to "go and reevaluate by including limits from geology, geography, and

rates of deployment as things are overestimated" – but how is this new? Except it has been somewhat quantified but again with many heterogeneous assumptions.

Minor Comments

1- I'm not sure if those bulk citations are useful for the readers. I encourage the author to do a better job in citing other works, keeping those that are most related to a claim.

2- In L131, "modelling framework", what framework? Elaborate more on "modeling".

3- All figure labels are small and hard (impossible) to read on print. There is no consistency in the font and style of figures. Each seems to be from a different source. Some have a background, closed box, some are white open-box, some have bold fonts, some none, some have very large fonts, some very tiny. All the figures should be updated to have more or less the same style with a font comparable to the body text.

4- In L163, should readers at this point know what a "growth modeling framework" is?

5- For Fig 2, I would add an x-axis to the bar chart too. Is the legend the same for both plots?

6- In L361, equations cannot be referred to before their appearance. Rephrase that line as "A description of the cumulative storage, $P(t)$ [GtCO₂], and storage rate, $Q(t)$ [GtCO₂ yr⁻¹], of CO₂ at time, t [yr], is outlined as [then write the equations here...]"

7- In L371, "Equation 3" is not required here.

8- In L437, "we compute 1000 random iterations" – why is 1000 supposed to be sufficient?

Reviewer #3 (Remarks to the Author):

This is a very good paper, offering a realistic perspective on the potential for CCS growth by the mid-century. Methods are clear and also replicable. The graphics are good and the conclusions are supported by data. However, I believe there are a few fundamental limitations in this study due to which I do not think it meets the stringent review criteria of Nature Communications. I do think that this paper would make a very strong case for a more specialized journal in the field of CCS. Please see my comments below:

1. The most important gap that the authors do not seem to have identified is the subject of injectivity. Even if there is a lot of storage potential available, it might not be utilized to the full extent in a short span of time because reservoir pressure will increase beyond the limit and may lead to fracture. This has been studied at a systems scale but this limitation is not addressed in the paper. Instead, the authors have used the growth curves associated with the hydrocarbon industry. While

relevant, I do not see this as an exact analogue. Particularly, countries may not unilaterally produce huge amounts of oil far exceeding demand, as that would lead to price falls. There is, however, no such constraint in terms of geologic CO₂ storage.

2. Authors also misidentify a gap in IAMs, “Crucially, CO₂ storage resources are not represented as an exhaustible resource within IAMs that otherwise characterise resources such as oil, gas, coal, and uranium using cumulative resource supply curves and decline rates (Line 100). I do not believe this is an accurate characterization. GCAM – for instance – has CCS supply curves that are inbuilt within the model https://jgcri.github.io/gcam-doc/v3.2/The_Energy_System. As a result, the authors misstate the extent of novelty of the current manuscript, in my view.

3. Similar efforts have also been published in the past seeking to align IAM projections with storage potential constraints <https://doi.org/10.1016/j.apenergy.2016.11.117>. As the authors have not cited this work and their novelty over it, I am concerned if the authors might have overlooked it.

4. Something else the authors likely overlook is the discourse and current trends around international CO₂ transport, as announced by countries like South Korea and Japan. These are part of well-announced strategies but not modeled here. See summary of such efforts here <https://doi.org/10.1016/j.rcradv.2023.200174>. This would then account for the issue mentioned in lines 204-212 regarding the mismatch between sources and sinks.

5. The fact that CCS required in 1.5-2C scenarios exceed government stated targets is not surprising. There is considerable literature out there outlining this <https://doi.org/10.1038/s41558-022-01508-0>

6. Authors have cited ref. 8 (i.e., AR6 database) on the range of geologic CO₂ projected to be stored by 2050 (Line 79). This may be accurate but my suggestion is to instead quote the interquartile range here. Some extreme scenarios on both ends might yield results that end up in such a wide range. If the interquartile range is quoted instead (as was the practice in many chapters of AR6), it will likely yield a more practical range. This gets reflected in results also. For instance, authors state that 6% of the scenarios are infeasible (lines 166-168). These would likely fall outside the interquartile range, meaning that the conclusion of this paper and the IAM ensemble is similar.

7. Also, in the context of the AR6 database, authors have not drawn sufficient distinction between 1.5C scenario with no or limited overshoot (C1), 1.5C w overshoot (C2), 2C and 2C w policy delay. It would have been really interesting to see how the delay/overshoot scenarios interface with the findings here. That again is a major gap here.

8. Line 83: I am having trouble understanding the comparison of 4 GtCO₂ given here. A more suitable comparison might be to cite this paper which essentially says that ~500 MtCO₂ has been stored so far since the 1970s combining EOR and dedicated storage
<https://doi.org/10.1016/j.oneear.2021.10.002>

REVIEWER COMMENTS

Reviewer #1 (Remarks to the Author):

Article presents interesting results in making projections of CO₂ storage by mid-century. It discusses the feasibility of scaling up carbon storage in subsurface reservoirs to mitigate climate change, as projected by the AR6. While the AR6 estimates a wide range of potential storage rates by 2050, this study highlights the importance of considering geological, geographical, and techno-economic limitations in these projections. Here are some comments that might improve the article.

Comments:

1) Line 81-83 - What does this value mean (production of CO₂ during oil and gas production?)

We are making a comparison of mass flows between the current oil and gas industry, and what is envisaged for CO₂ storage. We have modified the sentence for clarity:

Line 80: "The envisaged CO₂ storage industry is comparable to the current scale of the hydrocarbon industry. Globally, 4 Gt of oil was produced annually between 2011-2021¹⁴."

2) Line 94: some visual representation of the deviation of practical results with projections could be useful.

We have expanded Figure 3 to further illustrate the disparities between the outcomes of our modelling and the projections outlined across four climate categories of the AR6. We now show distinct box plots for the ranges of projections from the AR6 for the four groupings limiting warming to less than 2 °C.

3) Line 98 - Could you please provide the reasons, why these projects were not implemented? How these factors can be taken into account while building projection models? Which factors have the highest impact?

The success and failure of CCS projects in the USA was examined by Abdulla et al. (2020). They found that there are diverse attributes across engineering, economics, finance, and political economy impacting the likelihood of project success. These attributes include public opposition, plant siting, technology readiness, capital cost, credibility of incentives, and population proximity.

The modelling framework we are using aggregates limitations to growth arising from physical, engineering, economic, regulatory, and other areas into a simple decline in the rate of growth below exponential. There is no explicit representation of the individual risks identified in Abdulla et al. (2020), but the range of parameter space we consider within our modelling framework encompasses the uncertainties described therein. In the context of the present work, limitations arising from policy and economic issues manifest in the range of growth rate considered, from 10% – 20% annual growth (except for China). Limitations arising from the

geological system are encompassed within the uncertainty of the resource base, that we vary an order of magnitude in each direction around the central estimates.

We have added the following to address these points:

Line 436: “However, uncertainty ranging between 1-2 orders of magnitude within storage resource estimates persists for saline aquifers^{26,88,89,90}. This geological uncertainty can only be significantly reduced with further project development, which is essential to provide more detailed reservoir characterisation, and to reduce uncertainties arising from engineering aspects of the project design. Geological and engineering characteristics combine to govern the efficiency of the total pore volume use to store CO₂^{84,85,89}. Consequently, estimates of storage efficiency range from 0.5% - 4%⁹⁰. We note that many of the resource base assessments incorporate dynamic simulations into their estimates of the storage efficiency, and there is not a simple systematic relationship between dynamic and volumetric estimates of a storage resource. In light of this, we follow the approach established by Zhang et al.⁵⁶, and analyse scale-up trajectories with a storage resource base that ranges from a hypothetical minimum, 10 times lower than the central estimates of the resource base, and a hypothetical maximum, which is 10 times greater than the central estimate. In this way, our analysis implicitly incorporates uncertainties associated with the geological system, like the emergence of a leading limitation arising from injectivity, by the consideration of a far more restricted resource base than is currently assessed.”

Line490: “Comparisons of scenarios reflect the various drivers and limits to the modelled storage rate. Comparisons of modelled scenarios to the projections on storage rate from IAMs illustrate the range of technical feasibility. These scenarios are not predictive but provides an insight for the necessary market conditions, opportunities, and bottlenecks of what future long-term CO₂ storage development has envisioned. The Reference scenario sets out a range of storage rate within the maximum bound of feasible growth rate (i.e., 20% or 25%) and reflects broad differences of modelled storage rate across geographies based on central estimates of available storage resource base for each region. The Minimum scenario reflects a conservative case on modelled storage rate, limiting growth rate to a maximum of 10% and a storage resource base of only 10% of central estimates (hypothetical minimum). The Growth10% scenario explores the extent of which modelled storage rate is limited by growth only compared to the Reference and Minimum scenarios.”

We highlight the risks and uncertainties that have been examined in the following:

Line 91: “Globally, approximately 70% of the 149 projects proposed to be operational by 2020, aiming to store 130 Mt of CO₂ annually, were not implemented³⁷. Project cost, low technology readiness levels among the capture technology, and a lack of revenue streams, e.g., oil production, are among the main contributors to projects stopping^{37,38}. Among existing actively operational CCS projects, only around 9 Mt yr⁻¹ of a total capture capacity of 45 Mt yr⁻¹ is injected for dedicated storage, with the rest used for enhanced oil recovery³⁹. This discrepancy between real-world development and projected trajectories highlights limitations to CO₂ storage scale-up that are not captured by existing IAMs.”

4) Line 123-124 - can your projections include some practical, theoretical base (some coefficients that accounts trapping mechanisms, experimental studies - focusing on these mechanisms, hydrodynamic studies - to validate your projections? Probably, you can include some literature review of some alternatives for your approach. Is there some other research groups that are performing similar studies? what are the strong sides of your approach, and drawbacks? Adding a highlight of the novelty and global contribution in the

Main part would be beneficial.

Various studies have used simplified physics models focusing on limitations to CCS scale-up arising from either injectivity or plume migration (De Simone & Krevor, 2020; Szulczewski et al., 2012; Mathias et al. 2009). An alternative that has been developed are empirical models based entirely on historical analogues. Ringrose and Meckel (2019) utilised this method to evaluate trajectories for CO₂ storage scale-up regionally and globally. Grant et al. (2022), incorporated the maximum achievable extraction rate from the hydrocarbon sector as a dynamic constraint within a global energy system model to estimate CCS deployment. Their study indicates that CCS deployment might face greater challenges in China compared to North America and Europe, where a well-established historical hydrocarbon industry exists. This aligns well with our own findings, suggesting that the projections in AR6 overestimate the feasibility of contributions in China. Furthermore, their analysis also identifies a similar global injection scenario to our own analysis, estimating around 5 GtCO₂ yr⁻¹ by 2050, with the USA contributing over 1 GtCO₂ yr⁻¹.

We have modified the text to summarise place our work in the context of these approaches:

Line 99: “Many potentially leading geographic, geologic, and techno-economic limitations to subsurface resource exploitation are not yet represented in IAMs^{16,29,40,41,42,43,44}. Integrated assessment models use a range of constraints on the scale-up of CO₂ storage, including single values of global storage potential, supply cost curve, limits on injection rates, but some also have no subsurface-specific constraints^{1,2,4,7,8,9,11,12,13}. However, these constraints have no direct correlation to storage resource deployment in the resulting trajectories (Figure 1).

The evaluation of CO₂ storage scale-up by direct comparison to industrial analogues and using more restrictive storage capacities reveals significant global and regional discrepancies from the projections of conventional IAMs^{29,45,46,47}. The use of historical hydrocarbon production rates, and the drilling of wells, as a proxy for CO₂ storage scale-up shows that historical rates of engineering are in line with projections in the USA, whereas there is less precedent in Asian countries, and particularly China^{29,46,47}. Embedding these empirical restrictions into IAMs significantly restricts the deployment of CCS^{29,45}. However, this direct use of historical analogue data is difficult to generalise as a modelling approach because of challenges in translating the original datasets to units and processes associated with subsurface CO₂ storage, and the limited flexibility in the exploration of uncertainties in deployment trajectories.”

Line 115: “Growth modelling frameworks, of which the logistic curve is the most widely used mathematical form, have been developed expressly to create future projections of natural resource consumption, based on observed patterns of growth in extractive industries⁴⁸⁻⁵⁴. The application of a growth model to the scale-up of CO₂ storage globally, without resolving regional variations, suggested deployment of no more than 11 Gt by 2050, but with many hundreds of Gt potentially stored by the end of the century⁵⁵. This approach has also been used at the regional scale, identifying boundaries for plausible scale-up trajectories in the USA and Europe^{55,57}. The strength of the logistic model is derived from a combination of its simplicity, its embedding of the impact of the depletable nature of CO₂ storage resources on growth, and the ability to make use of growth parameterisations based on historical analogues from extractive and other industries⁵⁸. The important trade-offs come from its lack of granularity. The framework does not explicitly define particular incentives or inhibitions to growth arising from engineering, geology, and economics, and it is only valid for evaluating growth of a large number of deployments over at least multiple decades^{48,59-61}.”

Line 306: “These findings are remarkably consistent with the analyses that impose empirical constraints directly from historical oil and gas production, and also find a global annual injection rate of CO₂ limited to 5-6 Gt yr⁻¹, with the USA as the largest contributor to that total²⁹. The convergence of these estimates from distinct modelling approaches suggests that a benchmark for the upper bounds of the feasible global deployment of CCS by 2050 could be set at 5-6 Gt yr⁻¹.”

Line 380: “We make use of a logistic growth model which is an empirical mathematical framework that has been applied to describe growth in the extractive industries, including oil and gas, coal, and uranium production^{48,49,50,51,52,53,74}. Over time, the model’s application has been expanding beyond fossil fuel resources with several studies demonstrating the suitability of logistic curves to model energy production from renewable sources, nuclear, and the rates of technological substitution⁷⁵⁻⁸⁰.”

Line 436: “However, uncertainty ranging between 1-2 orders of magnitude within storage resource estimates persists for saline aquifers^{26,88,89,90}. This geological uncertainty can only be significantly reduced with further project development, which is essential to provide more detailed reservoir characterisation, and to reduce uncertainties arising from engineering aspects of the project design. Geological and engineering characteristics combine to govern the efficiency of the total pore volume use to store CO₂^{84,85,89}. Consequently, estimates of storage efficiency range from 0.5% - 4%⁹⁰. We note that many of the resource base assessments incorporate dynamic simulations into their estimates of the storage efficiency, and there is not a simple systematic relationship between dynamic and volumetric estimates of a storage resource. In light of this, we follow the approach established by Zhang et al.⁵⁶, and analyse scale-up trajectories with a storage resource base that ranges from a hypothetical minimum, 10 times lower than the central estimates of the resource base, and a hypothetical maximum, which is 10 times greater than the central estimate. In this way, our analysis implicitly incorporates uncertainties associated with the geological system, like the emergence of a leading limitation arising from injectivity, by the consideration of a far more restricted resource base than is currently assessed.”

5) Figure 3 must be fixed - legends should be crossed by the zoomed part of the Figure (exceeding 3.6 Gt). Probably, it should be divided into a)b)c)

We have improved the visualisation of Figure 3, differentiating the graphs into three subplots and placed the legend accordingly over plots A and B for the reference scenario.

6) As authors mentioned, the capacity of potential CO₂ storage is indefinitely large value, however, not all the reservoirs/aquifers can be used for potential CO₂ storage (Line 392). Authors used reports with CO₂ storage estimates, however, it is not clear, how the selection criteria (geological and engineering constraints) of existing resources was applied in the projections. In line 407 Authors mention, that reservoir characteristics were used to test sensitivity analysis, how these results were used in the projections? Probably the description of the methodology could be extended. Also, all the constraints can be given in the Table for better visual understanding.

Zahasky & Krevor (2020) assessed the uncertainty in storage resource assessments carried out using different assessment methodologies, including volumetric and dynamic (injectivity-limited) approaches, across resources ranging from individual fields to basin scales. For a given storage resource with multiple

assessments, estimates vary 1 to 2 orders of magnitude. Different assessment techniques impose varying degrees of constraints; for instance, dynamic studies account for factors determining resource accessibility, such as engineering limitations and geological complexities arising from faults and reservoir overpressurization (Teletzke et al., 2018), while volumetric estimates apply an efficiency factor to the volume of the pore space. Storage efficiency factors, which are also sometimes derived from dynamic studies, vary between 0.5% to 4%, and give rise to the overall uncertainty in the storage resource estimate (Bachu et al., 2007).

These findings underpin our approach of including the simple order of magnitude sensitivity to account for uncertainty in the storage resource base, i.e., with scenarios where the storage resource base is constrained to 10% of central estimates.

We have modified Table 2 and Figure 6 to provide an improved visualisation of constraints used in the logistic model.

These points are emphasised in the following:

Line 436: “However, uncertainty ranging between 1-2 orders of magnitude within storage resource estimates persists for saline aquifers^{26,88,89,90}. This geological uncertainty can only be significantly reduced with further project development, which is essential to provide more detailed reservoir characterisation, and to reduce uncertainties arising from engineering aspects of the project design. Geological and engineering characteristics combine to govern the efficiency of the total pore volume use to store CO₂^{84,85,89}. Consequently, estimates of storage efficiency range from 0.5% - 4%⁹⁰. We note that many of the resource base assessments incorporate dynamic simulations into their estimates of the storage efficiency, and there is not a simple systematic relationship between dynamic and volumetric estimates of a storage resource. In light of this, we follow the approach established by Zhang et al.⁵⁶, and analyse scale-up trajectories with a storage resource base that ranges from a hypothetical minimum, 10 times lower than the central estimates of the resource base, and a hypothetical maximum, which is 10 times greater than the central estimate. In this way, our analysis implicitly incorporates uncertainties associated with the geological system, like the emergence of a leading limitation arising from injectivity, by the consideration of a far more restricted resource base than is currently assessed.”

7) Conclusion might be enhanced by highlighting the contribution of this results for the industry, how it can affect the future of the industry, climate and human existence in general.

The Discussion and Conclusions section (from line 286) has been heavily modified to address this comment.

From line 318: “Projections from IAMs should revisit and add to the constraints that reflect limitations arising from subsurface storage development. Our results suggest that gigatonne scale mitigation from CO₂ storage is feasible, but with deployment trajectories and geographies differing from current projections emergent from IAMs. The Sixth Assessment Report includes unrealistic projections of CCS deployment, particularly in Asian countries, and these are widely used as benchmarks for progress measurement. This work demonstrates the practicality of using available data, subsurface storage resources, and a logistic growth modelling framework to provide multi-decadal constraints on the pace of development of CO₂ storage. Each modelled trajectory is anchored by estimates of region-

specific cumulative storage achieved by 2030, and the storage resource base for that region. Growth models do not have high temporal fidelity, they represent a statistical averaging over yearly fluctuations in development, but their simple parameterisation permits the rapid assessment of the impacts of uncertainties in the capacity to realise planned developments, to incentivise growth, and in the understanding of the underlying storage resource base.”

Reviewer #2 (Remarks to the Author):

Dear Editor,

This article investigates the feasibility of scaling up CO₂ storage considering geology and growth rates. Results suggest a maximum global storage rate of 16 GtCO₂ yr⁻¹ by 2050, with the USA contributing 60%. However, if constrained by government roadmaps, this drops to 5 GtCO₂ yr⁻¹, mainly due to limited deployment in the USA. Previous estimates overestimated storage in China, Indonesia, and South Korea. Achieving gigatonne scale mitigation requires updating projections to include geological and deployment rate limits.

The work can make valuable contributions to the fields. However, I have several comments and concerns outlined below that can further improve the quality of the paper prior to acceptance.

Major Comments

1- Technical uncertainty and risk mitigation play a significant role in the estimated value. Authors are encouraged to consider those factors in their analysis. For a comprehensive list of uncertainties and risks, see the review paper recently published (<https://doi.org/10.1016/j.igsce.2023.205117>). Without this being incorporated into the analysis, the numbers seem again overestimated (or underestimated), which weakens the contribution of this work.

The modelling framework we are using aggregates limitations to growth arising from physical, engineering, economic, regulatory, and other areas into a simple decline in the rate of growth below exponential. There is no explicit representation of the individual risks identified in Mahjour and Faroughi (2023), but the range of parameter space we consider within our modelling framework encompasses the uncertainties described therein. In the context of the present work, limitations arising from policy and economic issues manifest in the range of growth rate considered, from 10% – 20% annual growth (except for China). Limitations arising from the geological system are encompassed within the uncertainty of the resource base that we vary an order of magnitude in each direction around the central estimates.

We highlight the risks and uncertainties that have been examined in the following:

Line 91: “Globally, approximately 70% of the 149 projects proposed to be operational by 2020, aiming to store 130 Mt of CO₂ annually, were not implemented³⁷. Project cost, low technology readiness levels among the capture technology, and a lack of revenue streams, e.g., oil production, are among the main contributors to projects stopping^{37,38}. Among existing actively operational CCS projects, only around 9 Mt yr⁻¹ of a

total capture capacity of 45 Mt yr⁻¹ is injected for dedicated storage, with the rest used for enhanced oil recovery³⁹. This discrepancy between real-world development and projected trajectories highlights limitations to CO₂ storage scale-up that are not captured by existing IAMs.”

2- Methods and assumptions are provided after the results and conclusions. I understand why that can be the case, but a summary of those assumptions and modeling approaches (a few lines) with proper citation and reference to the appendix must be given before discussing the results, as results make no sense to readers until they know all the assumptions made in the analysis. A slight arrangement or proper definition of models and assumptions at the end of the introduction is required.

We have now included a summary of the methods in the main text prior to the discussion of results to facilitate clarity on the approach. This is provided in the following:

Line 128: “In this Article, we generate projections of CO₂ storage deployment geospatially around the world that are constrained by current or planned deployment, rates of growth, and the assessed geological resource base. We use a growth modelling framework employing a symmetric logistic curve, building directly on the work analyses of subsurface CO₂ storage in the USA and Europe from Zhang et al.⁵⁶ and Zhang et al.⁵⁷. We expand this to geographically resolved models of scale-up across North America, Europe, the Middle East, and Asia. We account for current CCS deployment and the geological constraints in each region by anchoring the modelled trajectories using the cumulative storage anticipated by 2030, and the assessed resource base. We formulate six scenarios under this framework, introducing variations in storage resource constraints, upper bounds on growth rates, and targeted regional limits on storage rate, to examine the uncertainty and possibilities surrounding projected CO₂ storage development (see Table 1). We model a distribution of storage rates for each region by selecting trajectories randomly within a bounding parameter space of feasible annual growth rates and the assessed storage resource base. The distribution of global storage rates is then the sum of the modelled storage contributions from each region. We use these projections of to identify a geography of feasible CO₂ storage scale-up, to analyse the trajectories in IAMs, and identify the impacts of regional contributions in driving global storage rates.”

3- With all these heterogeneous assumptions throughout the model and scenario development, how is the output of this paper any more statistically significant than simple range estimation by AR6? To be honest, I read this paper twice and was not convinced if the new modeling approach is any better or if it provided any new insights that are not already known in the field. For example, the claim “if constrained by government policies, the storage rate drops”, this is common sense! I encourage authors to morph the analysis toward revealing some aspects, like implications, that could enhance the field and better inform policymakers. Based on the abstract, the authors simply concluded for the readers to "go and reevaluate by including limits from geology, geography, and rates of deployment as things are overestimated" – but how is this new? Except it has been somewhat quantified but again with many heterogeneous assumptions.

The application of a growth model to both generate growth trajectories and analyse projections that have been created by other means is novel and leads

directly to the main advances in this work. As now summarised in the introductory section and in Figure 1, integrated assessment models use either a granular “bottom-up” representations of CO₂ storage, where annual deployment is determined by a cost-supply curve, or simple end-point restrictions on deployment like a maximal injection rate or cumulative storage resource use. These approaches are leading to infeasibly high projections when compared to relevant historical benchmarks, like the well construction or oil production.

Also now summarised in the introductory section is an alternative approach whereby historical empirical data from industrial analogues, like the production of oil and gas, is used directly to estimate scale-up rates or to constrain scale-up in integrated assessment models. The challenge with this is it is not easily generalised – there are difficulties and arbitrary decision making in translating units from analogues to CO₂ storage, and limited flexibility in exploring uncertainty around the empirical time-series data.

Our use of the growth model combines the empiricism of the historical analogues (e.g., the pattern of growth represented by the logistic curve, and the rate constraint), with the flexibility of the deterministic models to explore ranges of outcomes that may arise from the considerable uncertainties. In this way, we mitigate the inflexibility and problems associated with the translation of units inherent in using a time-series dataset from an industrial analogue like oil production. In other words, we can explore a range of parameter space by simply varying the initial growth rate and storage resource base in our modelling framework. At the same time, we avoid the lack of constraints that are leading the cost-supply curve approaches to generate extraordinarily high rates of scale-up, when considered over multiple decades.

This combined flexibility and constraint has allowed us to apply the models with a regional geographic resolution. The conclusions regarding the impact of government constraints in the USA are notable; not because constraints lead to reductions in deployment, which as the reviewer points out is a trivial finding. Rather, we identify that the geography of CCS deployment in the AR6 trajectories is unrealistic when global deployment exceeds 6 Gt yr⁻¹. The IAMs are only able to achieve these scales of deployment with extraordinary scales of deployment in China, Indonesia, and South Korea. Deployment in the USA (and Europe) in the IAMs is relatively modest and not far in excess of government roadmaps. We would have reached the same conclusion if we had used the average deployment in the USA arising from the AR6 IAMs as our constraint.

We provide greater clarity around the context and contribution of this modelling approach in the introductory section:

Line 99: “Many potentially leading geographic, geologic, and techno-economic limitations to subsurface resource exploitation are not yet represented in IAMs^{16,29,40,41,42,43,44}. Integrated assessment models use a range of constraints on the scale-up of CO₂ storage, including single values of global storage potential, supply cost curve, limits on injection rates, but some also have no subsurface-specific constraints^{1,2,4,7,8,9,11,12,13}. However, these constraints have no direct correlation to storage resource deployment in the resulting trajectories (Figure 1).

The evaluation of CO₂ storage scale-up by direct comparison to industrial analogues and using more restrictive storage capacities reveals significant global and regional discrepancies from the projections of conventional IAMs^{29,45,46,47}. The use of historical hydrocarbon production rates, and the drilling of wells, as a proxy for CO₂ storage scale-up shows that historical rates of engineering are in line with projections in the USA, whereas there is less precedent in Asian countries, and particularly China^{29,46,47}. Embedding these empirical restrictions into IAMs significantly restricts the deployment of CCS^{29,45}. However, this direct use of historical analogue data is difficult to generalise as a modelling approach because of challenges in translating the original datasets to units and processes associated with subsurface CO₂ storage, and the limited flexibility in the exploration of uncertainties in deployment trajectories.

Growth modelling frameworks, of which the logistic curve is the most widely used mathematical form, have been developed expressly to create future projections of natural resource consumption, based on observed patterns of growth in extractive industries⁴⁸⁻⁵⁴. The application of a growth model to the scale-up of CO₂ storage globally, without resolving regional variations, suggested deployment of no more than 11 Gt by 2050, but with many hundreds of Gt potentially stored by the end of the century⁵⁵. This approach has also been used at the regional scale, identifying boundaries for plausible scale-up trajectories in the USA and Europe^{55,57}. The strength of the logistic model is derived from a combination of its simplicity, its embedding of the impact of the depletable nature of CO₂ storage resources on growth, and the ability to make use of growth parameterisations based on historical analogues from extractive and other industries⁵⁸. The important trade-offs come from its lack of granularity. The framework does not explicitly define particular incentives or inhibitions to growth arising from engineering, geology, and economics, and it is only valid for evaluating growth of a large number of deployments over at least multiple decades^{48,59-61}.”

The implications of the key findings are refined in the discussion:

Line 296: “The feasible regional distribution of CO₂ storage deployment varies considerably from the projections emergent from IAMs. The maximum rate of storage achieved globally is largely controlled by deployment in six regions, the USA, China, the UK, EU, Canada, and the Middle East, in order of decreasing impact. Projections from IAMs envision global mitigation from CCS by mid-century in excess of 10 Gt yr⁻¹ with relatively modest contributions from the USA. However, these projections vastly overestimate feasible deployment in China, Indonesia, and South Korea. As described above, when limiting 2050 deployment in the USA to 1 Gt yr⁻¹, as per projections in both government roadmaps and from IAMs, the global rate achieved in 2050 cannot exceed 6 Gt yr⁻¹. This further reduces to 5 Gt yr⁻¹ when government projections are also used for the EU (0.33 Gt yr⁻¹), and UK (0.175 Gt yr⁻¹), and deployment in China is limited to, 0.216 Gt yr⁻¹, the volume equivalent of historical oil production¹⁴.”

Minor Comments

1- I'm not sure if those bulk citations are useful for the readers. I encourage the author to do a better job in citing other works, keeping those that are most related to a claim.

We have revised the citations. We limit the bulk citations to particular instances where we wish to convey the breadth of support for a statement, e.g., line 74 “... mitigation scenarios ... consistently anticipate widespread ... CO₂ storage.” We also work to associate particular studies with particular findings, even if those studies have also been earlier cited in a bulk citation. In other words, it should be

clear in all cases which studies are providing the evidence base for the preceding statement.

2- In L131, “modelling framework”, what framework? Elaborate more on “modeling”.

We have revised the following to clarify the modelling approach:

Line 115: “Growth modelling frameworks, of which the logistic curve is the most widely used mathematical form, have been developed expressly to project natural resource consumption, based on observed patterns of growth in extractive industries⁴⁸⁻⁵⁴.”

3-All figure labels are small and hard (impossible) to read on print. There is no consistency in the font and style of figures. Each seems to be from a different source. Some have a background, closed box, some are white open-box, some have bold fonts, some none, some have very large fonts, some very tiny. All the figures should be updated to have more or less the same style with a font comparable to the body text.

We have increased the font size of all figure labels and ensured consistency in font and style has been applied across all figures. An exception is that the quantitative axes labels on Figure 2 are too small to read; however, we have emphasised that Figure 2 is a schematic to illustrate that we have generated such graphs as the basis of our results. The full graphs are provided as a part of the supplementary information. This is also explained in the caption for the figure.

4- In L163, should readers at this point know what a “growth modeling framework” is?

We have now clarified this early in the manuscript at the following points:

Line 115: “Growth modelling frameworks, of which the logistic curve is the most widely used mathematical form, have been developed expressly to project natural resource consumption, based on observed patterns of growth in extractive industries⁴⁸⁻⁵⁴.”

Line 130: “We use a growth modelling framework employing a symmetric logistic curve, building directly on the work analyses of subsurface CO₂ storage in the USA and Europe from Zhang et al.⁵⁶ and Zhang et al.⁵⁷.”

5- For Fig 2, I would add an x-axis to the bar chart too. Is the legend the same for both plots?

We have now added the axis as suggested in the now Figure 3. The legend covers both the lower and upper plots (making use of the same horizontal axes), and the colour is consistent between the two, i.e., indicating the Modelled Scenarios in the upper plot.

6- In L361, equations cannot be referred to before their appearance. Rephrase that line as “A description of the cumulative storage, $P(t)$ [GtCO₂], and storage rate, $Q(t)$ [GtCO₂ yr⁻¹], of CO₂ at time, t [yr], is outlined as [then write the

equations here...]”.

This has been revised.

7- In L371, “Equation 3” is not required here.

Equation 3 and the associated text has been removed.

8- In L437, “we compute 1000 random iterations” – why is 1000 supposed to be sufficient?

We are generating random distributions. From Figure 3 you can observe that the distributions are normal to log-normal distributions. One thousand samples are well in excess of what is required to generate such distributions, i.e., the expectation values of the parameters (mean and standard deviation) of a log-normal distribution will converge with a sample size greater than around 150.

Reviewer #3 (Remarks to the Author):

This is a very good paper, offering a realistic perspective on the potential for CCS growth by the mid-century. Methods are clear and also replicable. The graphics are good and the conclusions are supported by data. However, I believe there are a few fundamental limitations in this study due to which I do not think it meets the stringent review criteria of Nature Communications. I do think that this paper would make a very strong case for a more specialized journal in the field of CCS. Please see my comments below:

1. The most important gap that the authors do not seem to have identified is the subject of injectivity. Even if there is a lot of storage potential available, it might not be utilized to the full extent in a short span of time because reservoir pressure will increase beyond the limit and may lead to fracture. This has been studied at a systems scale but this limitation is not addressed in the paper. Instead, the authors have used the growth curves associated with the hydrocarbon industry. While relevant, I do not see this as an exact analogue. Particularly, countries may not unilaterally produce huge amounts of oil far exceeding demand, as that would lead to price falls. There is, however, no such constraint in terms of geologic CO₂ storage.

The modelling framework we are using aggregates limitations to growth arising from physical, engineering, economic, regulatory, and other areas into a simple decline in the rate of growth below exponential. The maximum growth we allow, e.g., less than 20% sustained annual growth, is not just an analogue from the oil and gas industry, but a broadly emergent pattern from a range of extractive industries dependent on a natural resource base, including uranium extraction and coal production. These models, of which the logistic curve is most widely used, are parameterised by the total resource base in part because of an acknowledgement of the ways in which the physical limits of a finite resource can inhibit growth – in the present case, injectivity limitations would be reflected in the slowdown in growth, as ever more marginal resources are developed.

There is no explicit representation of injectivity, but the assumption that it would manifest as a limit on the growth rate, and a slow-down in growth as the total resource base is consumed. In the context of the present work, this and other potential limitations arising from the geological system are encompassed within the uncertainty of the resource base. We have observed that repeat resource assessments for geological CO₂ storage typically range 1-2 orders of magnitude (Zahasky and Krevor, 2020), due to variations in consideration, e.g., of pore volume versus injectivity. This makes up part of the rationale in the present study for considering scenarios with 10% of the currently assessed resource base.

We have modified the text to address the above points:

Line 380: “We make use of a logistic growth model which is an empirical mathematical framework that has been applied to describe growth in the extractive industries, including oil and gas, coal, and uranium production^{48,49,50,51,52,53,74}. Over time, the model’s application has been expanding beyond fossil fuel resources with several studies demonstrating the suitability of logistic curves to model energy production from renewable sources, nuclear, and the rates of technological substitution⁷⁵⁻⁸⁰.”

Line 436: “However, uncertainty ranging between 1-2 orders of magnitude within storage resource estimates persists for saline aquifers^{26,88,89,90}. This geological uncertainty can only be significantly reduced with further project development, which is essential to provide more detailed reservoir characterisation, and to reduce uncertainties arising from engineering aspects of the project design. Geological and engineering characteristics combine to govern the efficiency of the total pore volume use to store CO₂^{84,85,89}. Consequently, estimates of storage efficiency range from 0.5% - 4%⁹⁰. We note that many of the resource base assessments incorporate dynamic simulations into their estimates of the storage efficiency, and there is not a simple systematic relationship between dynamic and volumetric estimates of a storage resource. In light of this, we follow the approach established by Zhang et al.⁵⁶, and analyse scale-up trajectories with a storage resource base that ranges from a hypothetical minimum, 10 times lower than the central estimates of the resource base, and a hypothetical maximum, which is 10 times greater than the central estimate. In this way, our analysis implicitly incorporates uncertainties associated with the geological system, like the emergence of a leading limitation arising from injectivity, by the consideration of a far more restricted resource base than is currently assessed.”

2. Authors also misidentify a gap in IAMs, “Crucially, CO₂ storage resources are not represented as an exhaustible resource within IAMs that otherwise characterise resources such as oil, gas, coal, and uranium using cumulative resource supply curves and decline rates (Line 100). I do not believe this is an accurate characterization. GCAM – for instance – has CCS supply curves that are inbuilt within the model [https://jgcri.github.io/gcam-doc/v3.2/The Energy System](https://jgcri.github.io/gcam-doc/v3.2/The_Energy_System). As a result, the authors misstate the extent of novelty of the current manuscript, in my view.

We have strengthened this discussion with a new figure and discussion of the constraints applied to the projection of CO₂ storage deployment across influential Integrated Assessment Models (IAMs), including GCAM4.2, GCAM5.3, REMIND1.6, REMIND2.1, IMAGE3.0, WTICH, POLES, MESSAGEix, COFFEE, GEM-E3, and TIAM-ECN collectively representing approximately 82% of the synthesised mitigation pathways compiled in the AR6 for the four climate categories aimed at limiting warming to less than 1.5°C and 2°C. The models variously include an allowed storage potential, a limit on the annual injection rate, and a cost-supply curves. Some models do not have subsurface-specific constraints, and CO₂ storage is deployed as the parent technology is deployed.

There is no correlation in the maximum deployment rate and the ways in which CO₂ storage is constrained within the IAMs. For example, GCAM5.3 uses a cost supply curve, but produces the largest projected 2050 global CO₂ storage rate, 30 GtCO₂ yr⁻¹.

The novelty in our work derives from the use of the growth modelling framework, which is a development in capturing the benefits of the purely empirical approaches. Rather than beholden to a particular time-series of oil and gas production, we can extract patterns of growth from multiple potentially analogous industries and treat them as a range of justified projections. This removes the requirement for a number of arbitrary decisions, e.g., in how to translate oil production to CO₂ injection equivalent, and also bolsters the empirical support by

permitting consideration of growth trajectories from other industries with less easily translatable units of production, like uranium or coal production.

We have modified the referenced sentence and subsequent paragraph to:

Line 99: “Many potentially leading geographic, geologic, and techno-economic limitations to subsurface resource exploitation are not yet represented in IAMs^{16,29,40,41,42,43,44}. Integrated assessment models use a range of constraints on the scale-up of CO₂ storage, including single values of global storage potential, supply cost curve, limits on injection rates, but some also have no subsurface-specific constraints^{1,2,4,7,8,9,11,12,13}. However, these constraints have no direct correlation to storage resource deployment in the resulting trajectories (Figure 1).

The evaluation of CO₂ storage scale-up by direct comparison to industrial analogues and using more restrictive storage capacities reveals significant global and regional discrepancies from the projections of conventional IAMs^{29,45,46,47}. The use of historical hydrocarbon production rates, and the drilling of wells, as a proxy for CO₂ storage scale-up shows that historical rates of engineering are in line with projections in the USA, whereas there is less precedent in Asian countries, and particularly China^{29,46,47}. Embedding these empirical restrictions into IAMs significantly restricts the deployment of CCS^{29,45}.”

Line 115: “Growth modelling frameworks, of which the logistic curve is the most widely used mathematical form, have been developed expressly to project natural resource consumption, based on observed patterns of growth in extractive industries⁴⁸⁻⁵⁴.”

Line 121: “The strength of the logistic model is derived from a combination of its simplicity, its embedding of the impact of the depletable nature of CO₂ storage resources on growth, and the ability to make use of growth parameterisations based on historical analogues from extractive and other industries⁵⁸. The important trade-offs come from its lack of granularity. The framework does not explicitly define particular incentives or inhibitions to growth arising from engineering, geology, and economics, and it is only valid for evaluating growth of a large number of deployments over at least multiple decades^{48,59-61}.”

Line 128: “In this Article, we generate projections of CO₂ storage deployment geospatially around the world that are constrained by current or planned deployment, rates of growth, and the assessed geological resource base. We use a growth modelling framework employing a symmetric logistic curve, building directly on the work analyses of subsurface CO₂ storage in the USA and Europe from Zhang et al.⁵⁶ and Zhang et al.⁵⁷. We expand this to geographically resolved models of scale-up across North America, Europe, the Middle East, and Asia. We account for current CCS deployment and the geological constraints in each region by anchoring the modelled trajectories using the cumulative storage anticipated by 2030, and the assessed resource base. We formulate six scenarios under this framework, introducing variations in storage resource constraints, upper bounds on growth rates, and targeted regional limits on storage rate, to examine the uncertainty and possibilities surrounding projected CO₂ storage development (see Table 1). We model a distribution of storage rates for each region by selecting trajectories randomly within a bounding parameter space of feasible annual growth rates and the assessed storage resource base. The distribution of global storage rates is then the sum of the modelled storage contributions from each region. We use these projections of to identify a geography of feasible CO₂ storage scale-up, to analyse the trajectories in IAMs, and identify the impacts of regional contributions in driving global storage rates.”

We have now summarised the various constraints adopted across existing IAMs in Figure 1.

3. Similar efforts have also been published in the past seeking to align IAM projections with storage potential constraints <https://doi.org/10.1016/j.apenergy.2016.11.117>. As the authors have not cited this work and their novelty over it, I am concerned if the authors might have overlooked it.

We have now included this work in our main text:

Line 105: “The evaluation of CO₂ storage scale-up by direct comparison to industrial analogues and using more restrictive storage capacities reveals significant global and regional discrepancies from the projections of conventional integrated assessment models^{29,45,46,47}.”

Line 110: “Embedding these empirical restrictions into integrated assessment models significantly restricts the deployment of CCS^{29,45}.”

4. Something else the authors likely overlook is the discourse and current trends around international CO₂ transport, as announced by countries like South Korea and Japan. These are part of well-announced strategies but not modeled here. See summary of such efforts here <https://doi.org/10.1016/j.rcradv.2023.200174>. This would then account for the issue mentioned in lines 204-212 regarding the mismatch between sources and sinks.

The modelling in our work is not limited by source to sink matching of CO₂ chains, but rather is a representation of the limitations to growth arising from the development of the subsurface infrastructure, irrespective of the source of CO₂. We have modified the text and included the suggested reference:

Line 311: “While technically feasible, the range of projections between 6-16 GtCO₂ yr⁻¹, representing a large proportion of the AR6 pathways, are fundamentally more uncertain than lower projections because of the absence of any current business or political framework under which this scale of industry would operate. In these scenarios the USA would have to store a majority percentage of its current CO₂ emissions. International agreements for the disposal of CO₂ are emerging in Europe and Asia⁶⁴. However, plans that would incentivise such a large-scale deployment in North America are not currently in development.”

5. The fact that CCS required in 1.5-2C scenarios exceed government stated targets is not surprising. There is considerable literature out there outlining this <https://doi.org/10.1038/s41558-022-01508-0>

The novelty in our work is to identify trajectories that represent CO₂ storage deployment in a feasible manner. To that end, we provide simplified guidance that can be useful for constraining CCS deployment: that trajectories should not be considered where more than 16 GtCO₂ yr⁻¹ is deployed, and that trajectories between 6 – 16 GtCO₂⁻¹ are fundamentally more uncertain than those deploying less than 6 GtCO₂⁻¹ in 2050.

These points are most succinctly made in the first two paragraphs of the section Discussion and Conclusions.

Line 286: “We evaluate feasible global CO₂ storage rates for 2050 combining contributions across ten geographic regions which have currently active or planned CCS deployment. There is a maximum feasible combined CO₂ storage rate of 16 Gt yr⁻¹ by 2050, encompassing 92% of the 689 projections of scale-up in the 1.5 and 2 °C climate categories of the Sixth Assessment Report of the IPCC. The feasible limit reduces to 13 Gt yr⁻¹ for our reference scenario where the central estimates of storage resources are used. However, if deployment in the USA is limited to projections in government roadmaps, or indeed central projections from pathways in the AR6, the global deployment of CCS is further limited to no more than 6 Gt yr⁻¹. Finally, in the most conservative scenarios we explored, limiting the annual growth rate to less than 10% and the storage resource base to 10% of the current estimates reduces the attainable global storage rate to less than 1 Gt yr⁻¹.”

The feasible regional distribution of CO₂ storage deployment varies considerably from the projections emergent from IAMs. The maximum rate of storage achieved globally is largely controlled by deployment in six regions, the USA, China, the UK, EU, Canada, and the Middle East, in order of decreasing impact. Projections from IAMs envision global mitigation from CCS by mid-century in excess of 10 Gt yr⁻¹ with relatively modest contributions from the USA. However, these projections vastly overestimate feasible deployment in China, Indonesia, and South Korea. As described above, when limiting 2050 deployment in the USA to 1 Gt yr⁻¹, as per projections in both government roadmaps and from IAMs, the global rate achieved in 2050 cannot exceed 6 Gt yr⁻¹. This further reduces to 5 Gt yr⁻¹ when government projections are also used for the EU (0.33 Gt yr⁻¹), and UK (0.175 Gt yr⁻¹), and deployment in China is limited to, 0.216 Gt yr⁻¹, the volume equivalent of historical oil production¹⁴. These findings are remarkably consistent with the analyses that impose empirical constraints directly from historical oil and gas production, and also find a global annual injection rate of CO₂ limited to 5-6 Gt yr⁻¹, with the USA as the largest contributor to that total²⁹. The convergence of these estimates from distinct modelling approaches suggests that a benchmark for the upper bounds of the feasible global deployment of CCS by 2050 could be set at 5-6 Gt yr⁻¹.”

6. Authors have cited ref. 8 (i.e., AR6 database) on the range of geologic CO₂ projected to be stored by 2050 (Line 79). This may be accurate but my suggestion is to instead quote the interquartile range here. Some extreme scenarios on both ends might yield results that end up in such a wide range. If the interquartile range is quoted instead (as was the practice in many chapters of AR6), it will likely yield a more practical range. This gets reflected in results also. For instance, authors state that 6% of the scenarios are infeasible (lines 166-168). These would likely fall outside the interquartile range, meaning that the conclusion of this paper and the IAM ensemble is similar.

We have revised this and now quote the interquartile range alongside the full range of projected pathways synthesised in the AR6. Our analysis shows that around 8% of the AR6 pathways fall within the range that we identify as technically infeasible, and these are indeed outside of the interquartile range in each of the climate categories. Figure 3 highlights a significant proportion of the upper quartile, including both limited and high overshoot 1.5°C pathways and the 2°C (>67%) pathway, extending into the infeasible range.

Although feasible, a large part of the interquartile range falls within the bounds of our modelled scenario “Maximum”, requiring an average annual growth of 20% and a storage resource base potentially an order of magnitude higher than existing central estimates. These findings underscore the demanding market conditions inherent in these scenarios.

These points are highlighted in the following:

Line 76: “By mid-century, the inter-quartile range of annual injection rates of CO₂ in these scenarios is more than 6 GtCO₂ yr⁻¹ (the full range is 1-30 GtCO₂ yr⁻¹)³”
xf

Line 183: “The comparison reveals several limitations in the projections from IAMs. Around 8% of the trajectories from the AR6 (56 out of a total of 689 model outcomes) project rates of storage in 2050 that are greater than 16 Gt yr⁻¹. Our analysis identifies these pathways as infeasible, requiring sustained annual growth in excess of 20% and storage resources in excess of the theoretical maximum that has been evaluated for individual countries (see Maximum scenario, Table 1). The interquartile range of projected pathways for both 1.5 and 2 °C (>67% likelihood), and the IEA Net Zero Emission target⁵ – widely considered as a benchmark to decarbonise the global energy sector – are achievable by storage scale-up modelled largely in the Reference and the Maximum scenarios. In contrast, limiting sustained annual growth to <10%, a rate still greater what has been achieved in the past 20 years in the CCS industry³⁵, inhibits the attainable aggregate global storage rate to a maximum of 1 Gt yr⁻¹, below any projections of storage deployment in the 1.5 and 2 °C pathways of the AR6.”

7. Also, in the context of the AR6 database, authors have not drawn sufficient distinction between 1.5C scenario with no or limited overshoot (C1), 1.5C w overshoot (C2), 2C and 2C w policy delay. It would have been really interesting to see how the delay/overshoot scenarios interface with the findings here. That again is a major gap here.

Figure 3 has been updated to delineate the distinctions among the four climate categories (C1, C2, C3, and C4) outlined in the AR6, aimed at limiting global warming to less than 1.5°C and 2°C.

8. Line 83: I am having trouble understanding the comparison of 4 GtCO₂ given here. A more suitable comparison might be to cite this paper which essentially says that ~500 MtCO₂ has been stored so far since the 1970s combining EOR and dedicated storage <https://doi.org/10.1016/j.oneear.2021.10.002>

We have revised the sentence to:

Line 80: “The envisaged CO₂ storage industry is comparable to the current scale of the hydrocarbon industry. Globally, 4 Gt of oil was produced annually between 2011-2021¹⁴.”

References cited herein are provided in the reference list of main manuscript. We provide the reference number here: 29, 37, 38, 42, 43, 44, 46, 55, 83, and 89.

REVIEWER COMMENTS

Reviewer #1 (Remarks to the Author):

Thank you very much for the changes made. I would recommend some minor revisions.

Comments:

1) In Figure 1 there are some abbreviations used: REMIND, WITCH, GEM, TIAM and etc., but no description is given for these Figure. The description of these models comes only in the Figure capture. Please, check.

2) The work still did not include/mention any experimental studies performed to study the trapping mechanisms of CO₂, that potentially can validate limits from geology, geography, and rates of deployment.. This part should be extended.

3) Line 436-438 - "This geological uncertainty can only be significantly reduced with further project development, which is essential to provide more detailed reservoir characterisation, and to reduce uncertainties arising from engineering aspects of the project design." Here experimental and numerical studies can be mentioned - what kind of uncertainties can be reduced? How these experimental studies can be incorporated into dynamic simulations? can your models be compared to the results of some hydrodynamic models performed for some existing CO₂ storage field-cases

Reviewer #2 (Remarks to the Author):

The authors have carefully addressed my comments and concerns, and thus the paper can be accepted in its current form.

Reviewer #3 (Remarks to the Author):

The authors have addressed my comments from the previous round. Thanks for the elaboration.

REVIEWER COMMENTS

Reviewer #1 (Remarks to the Author):

Thank you very much for the changes made. I would recommend some minor revisions.

Comments:

1) In Figure 1 there are some abbreviations used: REMIND, WITCH, GEM, TIAM and etc., but no description is given for these Figure. The description of these models comes only in the Figure capture. Please, check.

We have improved the description of the models in Figure 1, and have modified the following to text to highlight the references where full documentations of the discussed IAMs can be found:

Line 102: "Integrated assessment models use a range of constraints on the scale-up of CO₂ storage (See Figure 1. For full descriptions of the integrated assessment models included in the figure, see ref. ^{1,2,4,7,8,9,11,12,13}). These include single values of global storage potential, supply cost curves, and limits on injection rates, but some also have no subsurface-specific constraints. However, these constraints have no direct correlation to storage resource deployment in the resulting trajectories. Figure 1 shows that the use of cost supply curves, the most granular representation of subsurface storage in integrated assessment models, can lead to both the largest and among the smallest scales of deployment in the projections."

2) The work still did not include/mention any experimental studies performed to study the trapping mechanisms of CO₂, that potentially can validate limits from geology, geography, and rates of deployment.. This part should be extended.

We have added a discussion and citations to experimental studies characterising flow physics (Akbarabadi & Piri, 2013; Ringrose et al., 2021; Krevor et al., 2015), and modelling studies that have used these properties in physics-based models to analyse limitations to CCS scale-up in specific regions arising from either injectivity or plume migration (Birkholzer & Zhou, 2009; De Simone & Krevor, 2020; Mathias et al. 2009; Szulczewski et al., 2012; Szulczewski et al., 2014; Zhou et al. 2009).

These changes have been made in the Main text:

Line 111: "The evaluation of CO₂ storage scale-up by using more restrictive storage capacities or by direct comparison to industrial analogues reveals significant global and regional discrepancies from the projections of conventional integrated assessment models^{29,42,43,44,45,46,47}. The fundamental flow physics of CO₂ migration and trapping have been characterised in laboratory studies^{48,49,50}. These properties have been incorporated into physics-based models to analyse limitations to large-scale CCS deployment arising from injectivity and plume migration in specific regions^{42,43,44,51,52,53}."

3) Line 436-438 - "This geological uncertainty can only be significantly reduced with further project development, which is essential to provide more detailed

reservoir characterisation, and to reduce uncertainties arising from engineering aspects of the project design." Here experimental and numerical studies can be mentioned - what kind of uncertainties can be reduced? How these experimental studies can be incorporated into dynamic simulations? can your models be compared to the results of some hydrodynamic models performed for some existing CO₂ storage field-cases

We have added the following to address this point:

Line 509: "Reservoir properties characterised in the laboratory and using geophysical techniques, combined with well hydraulic tests and observations of the movement of injected CO₂, form the empirical basis for simulations used to forecast and history match field-scale projects. These studies can reduce the geological and engineering uncertainty pertaining to the practical use of the CO₂ storage resource⁹⁷. Geological and engineering characteristics combine to govern the efficiency of the total pore volume use to store CO₂^{90,91,95}. As a result, estimates of storage efficiency range from 0.5% - 4%⁹⁶. Many resource base assessments use estimates of storage efficiency obtained using reservoir simulation^{87,88,98}, and there is not a simple systematic relationship between dynamic and volumetric estimates of a storage resource."

Line 369: "An obvious future development would be to combine the multi-decadal constraints from growth models with the more granular cost supply curves and physics-based models of injectivity that can provide deterministic estimates of yearly fluctuations in development."

References cited herein are provided in the reference list of main manuscript. We provide the reference number here: 42, 43, 44, 48, 49, 50, 51, 52, 53.